# MEAN-FIELD BEHAVIOUR OF NEURAL TANGENT KERNEL FOR DEEP NEURAL NETWORKS

## ABSTRACT

Recent work by Jacot et al. (2018) has showed that training a neural network of any kind with gradient descent in parameter space is equivalent to kernel gradient descent in function space with respect to the Neural Tangent Kernel (NTK). Lee et al. (2019) built on this result to show that the output of a neural network trained using full batch gradient descent can be approximated by a linear model for wide networks. In parallel, a recent line of studies (Schoenholz et al. (2017), Hayou et al. (2019)) suggested that a special initialization known as the Edge of Chaos leads to good performance. In this paper, we bridge the gap between these two concepts and show the impact of the initialization and the activation function on the NTK as the network depth becomes large. We provide experiments illustrating our theoretical results.

## 1 INTRODUCTION

Deep neural networks have achieved state-of-the-art results on numerous tasks; see, e.g., Nguyen & Hein (2018), Du et al. (2018b), Zhang et al. (2017). Although the loss function is not convex, Gradient Descent (GD) methods are often used successfully to learn these models. It has been actually recently shown that for certain overparameterized deep ReLU networks, GD converges to global minima ((Du et al., 2018a)). Similar results have been obtained for Stochastic Gradient Descent (SGD) ((Zou et al., 2018)).

The training dynamics of wide neural networks with GD is directly linked to kernel methods. Indeed, Jacot et al. (2018) showed that training a neural network with full batch GD in parameter space is equivalent to a functional GD i.e. a GD in a functional space with respect to a kernel called Neural Tangent Kernel (NTK). Du et al. (2019) used a similar approach to prove that full batch GD converges to global minima for shallow neural networks and Karakida et al. (2018) linked the Fisher Information Matrix to the NTK and studied its spectral distribution for infinite width networks. The infinite width limit for different architectures was studied by Yang (2019) who introduced a tensor formalism that can express most of the computations in neural networks. Lee et al. (2019) studied a linear approximation of the full batch GD dynamics based on the NTK and gave an method to approximate the NTK for different architectures. Finally, Arora et al. (2019) gives an efficient algorithm to compute exactly the NTK for convolutional architectures (Convolutional NTK or CNTK). In all of these papers, authors studied only the effect of infinite width on the NTK. The aim of this paper is to tackle the infinite depth limit.

In parallel, the impact of the initialization and activation function on the performance of wide deep neural networks has been studied in Hayou et al. (2019), Lee et al. (2018), Schoenholz et al. (2017), Yang & Schoenholz (2017). These works analyze the forward/backward propagation of some quantities through the network at the initial step as a function of the initial parameters and the activation function. They propose a set of parameters and activation functions so as to ensure a deep propagation of the information at initialization. While experimental results in these papers suggest that such selection also leads to overall better training procedures (i.e. beyond the initialization step), it remains unexplained why this is the case. In this paper, we link the initialization hyper-parameters and the activation function to the behaviour of the NTK which controls the training of DNNs, this could potentially explain the good performance. We provide a comprehensive study of the impact of the initialization and the activation function on the NTK and therefore on the resulting training dynamics for wide and deep networks. In particular, we show that an initialization known as the Edge

of Chaos (Yang & Schoenholz, 2017) leads to better training dynamics and that a class of smooth activation functions discussed in (Hayou et al., 2019) also improves the training dynamics compared to ReLU-like activation functions (see also Clevert et al. (2016)). We illustrate these theoretical results through simulations. All the proofs are detailed in the Supplementary Material which also includes additional theoretical and experimental results.

## 2 MOTIVATION AND RELATED WORK

### NEURAL TANGENT KERNEL

Recent work by Jacot et al. (2018) has shown that the training dynamics of neural networks are captured by the Neural Tangent Kernel (NTK). In the infinite width limit (wide neural networks), the NTK converges to a kernel that remains unchanged as the training time grows. While this is only true in the infinite width limit, Lee et al. (2018) showed that a first order linear approximation of the training dynamics (approximation of the NTK by its value at the initialization step) leads to comparable performances for different architectures. More recently, Bietti & Mairal (2019) studied the RKHS of the NTK for a two layers convolutional neural network with ReLU activation and provided a spectral decomposition of the kernel, while in Arora et al. (2019), the authors propose an algorithm to compute the NTK for convolutional neural networks. However, for finite width neural networks, Arora et al. (2019) observed a gap between the performances of the linear model derived from the NTK and the deep neural network, which is mainly due to the fact that the NTK changes with time. To fill this gap, Huang & Yau (2019) studied the dynamics of the NTK as a function of the training time for finite width neural networks and showed that the NTK dynamics follow an infinite hierarchy of ordinary differential equations baptised Neural Tangent Hierarchy (NTH). In this paper, we consider the limit of infinite width neural networks (mean-field approximation), and we study the behaviour of the NTK as the network depth goes to infinity assuming the width is infinite.

### EDGE OF CHAOS AND ACTIVATION FUNCTION

Recent works by Hayou et al. (2019) and Schoenholz et al. (2017) have shown that weight initialization plays a crucial role in the training speed of deep neural networks (DNNs). In Schoenholz et al. (2017), the authors demonstrate that only a special initialization can lead to good performance. This initialization is known as the 'Edge of Chaos' since it represents a transition between two phases : an ordered phase and a chaotic phase. When the DNN is initialized on the ordered phase, the output function of the DNN is constant almost everywhere, because the correlation of the outputs of two different inputs converges to 1 as the number of layers becomes large. On the other hand, when the DNN is initialized on the Chaotic phase, the output function is discontinuous almost everywhere as the depth goes to infinity. In this case, the correlation between the outputs of two different inputs converges to a value $c$ such that $|c| < 1$, therefore, very close inputs may lead to very different outputs. In Hayou et al. (2019), authors give a comprehensive analysis of the Edge of Chaos, and further show that a certain class of smooth activation functions outperform the ReLU-like activation functions in term of test accuracy on MNIST and CIFAR10.

### OUR CONTRIBUTIONS

In this paper, we bridge the gap between the two previous concepts of Neural Tangent Kernel and Edge of Chaos Initialization for DNNs. More precisely, we study the impact of the Edge of Chaos initialization and the activation function on the NTK as the depth $L$ goes to infinity. Our main results are :

1. With an Initialization on the ordered/chaotic phase, the NTK converges exponentially to a constant kernel with respect to the depth $L$, making the training impossible for DNNs (Lemma 1 and Proposition 1)

2. The Edge of Chaos initialization leads to an invertible NTK even in the infinite depth limit, making the model trainable even for very large depths (Proposition 2)

3. The Edge of Chaos initialization leads to a sub-exponential convergence rate of the NTK to the limiting NTK (w.r.t to $L$), which means that the 'information' carried by the NTK propagates deeper compared to an initialization on the ordered/chaotic phase (Propositon 2)

4. Using a certain class $\mathcal{S}$ of smooth activation functions can further slow this convergence, making this class of activation functions more suitable for DNNs

5. When adding Residual connections, we no longer need the initialisation on the Edge of Chaos, and the convergence of the NTK to the limiting NTK is always at a polynomial rate

# 3 NEURAL NETWORKS AND NEURAL TANGENT KERNEL

## 3.1 SETUP AND NOTATIONS

Consider a neural network model consisting of $L$ layers $(y^l)_{1 \leq l \leq L}$, with $y^l : \mathbb{R}^{n_{l-1}} \to \mathbb{R}^{n_l}$, $n_0 = d$ and let $\theta = (\theta^l)_{1 \leq l \leq L}$ be the flattened vector of weights and bias indexed by the layer's index and $p$ be the dimension of $\theta$. Recall that $\theta^l$ has dimension $n_l + 1$. The output $f$ of the neural network is given by some transformation $s : \mathbb{R}^{n_L} \to \mathbb{R}^o$ of the last layer $y^L(x)$; $o$ being the dimension of the output (e.g. number of classes for a classification problem). For any input $x \in \mathbb{R}^d$, we thus have $f(x, \theta) = s(y^L(x)) \in \mathbb{R}^o$. As we train the model, $\theta$ changes with time $t$ and we denote by $\theta_t$ the value of $\theta$ at time $t$ and $f_t(x) = f(x, \theta_t) = (f_j(x, \theta_t), j \leq o)$. Let $D = (x_i, y_i)_{1 \leq i \leq N}$ be the data set and let $\mathcal{X} = (x_i)_{1 \leq i \leq N}$, $\mathcal{Y} = (y_j)_{1 \leq j \leq N}$ be the matrices of input and output respectively, with dimension $d \times N$ and $o \times N$. For any function $g : \mathbb{R}^{d \times o} \to \mathbb{R}^k$, $k \geq 1$, we denote by $g(\mathcal{X}, \mathcal{Y})$ the matrix $(g(x_i, y_i))_{1 \leq i \leq N}$ of dimension $k \times N$.

Jacot et al. (2018) studied the behaviour of the output of the neural network as a function of the training time $t$ when the network is trained using a gradient descent algorithm. Lee et al. (2019) built on this result to linearize the training dynamics. We recall hereafter some of these results.

For a given $\theta$, the empirical loss is given by $\mathcal{L}(\theta) = \frac{1}{N} \sum_{i=1}^{N} \ell(f(x_i, \theta), y_i)$. The full batch GD algorithm is given by

$$\hat{\theta}_{t+1} = \hat{\theta}_t - \eta \nabla_\theta \mathcal{L}(\hat{\theta}_t) \tag{1}$$

where $\eta > 0$ is the learning rate.
Let $T > 0$ be the training time and $N_s = T/\eta$ be the number of steps of the discrete GD equation 1. The continuous time system equivalent to equation 1 with step $\Delta t = \eta$ is given by

$$d\theta_t = -\nabla_\theta \mathcal{L}(\theta_t) dt \tag{2}$$

This differs from the result by Lee et al. (2019) since we use a discretization step of $\Delta t = \eta$. It is well known that this discretization scheme leads to an error of order $\mathcal{O}(\eta)$ (see Appendix). As in Lee et al. (2019), Equation (2) can be re-written as

$$d\theta_t = -\frac{1}{N} \nabla_\theta f(\mathcal{X}, \theta_t)^T \nabla_z \ell(f(\mathcal{X}, \theta_t), \mathcal{Y}) dt$$

where $\nabla_\theta f(\mathcal{X}, \theta_t)$ is a matrix of dimension $oN \times p$ and $\nabla_z \ell(f(\mathcal{X}, \theta_t), \mathcal{Y})$ is the flattened vector of dimension $oN$ constructed from the concatenation of the vectors $\nabla_z \ell(z, y_i)_{|z=f(x_i, \theta_t)}, i \leq N$. As a result, the output function $f_t(x)$ satisfies the following ordinary differential equation

$$df_t(x) = \nabla_\theta f(x, \theta_t) d\theta_t = -\frac{1}{N} \nabla_\theta f(x, \theta_t) \nabla_\theta f(\mathcal{X}, \theta_t)^T \nabla_z \ell(f_t(\mathcal{X}), \mathcal{Y}) dt \in \mathbb{R}^o \tag{3}$$

The Neural Tangent Kernel (NTK) $K_\theta^L$ is defined as the $o \times o$ dimensional kernel satisfying: for all $x, x' \in \mathbb{R}^d$,

$$K_{\theta_t}^L(x, x') = \nabla_\theta f(x, \theta_t) \nabla_\theta f(x', \theta_t)^T \in \mathbb{R}^{o \times o}$$
$$= \sum_{l=1}^{L} \nabla_{\theta^l} f(x, \theta_t) \nabla_{\theta^l} f(x', \theta_t)^T. \tag{4}$$

We also define $K_{\theta_t}^L(\mathcal{X}, \mathcal{X})$ as the $oN \times oN$ matrix defined blockwise by

$$K_{\theta_t}^L(\mathcal{X}, \mathcal{X}) = \begin{pmatrix} K_{\theta_t}^L(x_1, x_1) & K_{\theta_t}^L(x_1, x_2) & \cdots & K_{\theta_t}^L(x_1, x_N) \\ K_{\theta_t}^L(x_2, x_1) & \cdots & \cdots & K_{\theta_t}^L(x_2, x_N) \\ \cdots & \cdots & \cdots & \cdots \\ K_{\theta_t}^L(x_N, x_1) & K_{\theta_t}^L(x_N, x_2) & \cdots & K_{\theta_t}^L(x_N, x_N) \end{pmatrix}$$

By applying equation 3 to the vector $\mathcal{X}$, one obtains

$$df_t(\mathcal{X}) = -\frac{1}{N}K_{\theta_t}^L(\mathcal{X}, \mathcal{X})\nabla_z\ell(f_t(\mathcal{X}), \mathcal{Y})dt, \tag{5}$$

meaning that for all $j \leq N$ $df_t(x_j) = -\frac{1}{N}K_{\theta_t}^L(x_j, \mathcal{X})\nabla_z\ell(f_t(\mathcal{X}), \mathcal{Y})dt$.

**Infinite width dynamics :** In the case of a fully connected feedforward neural network (FFNN) of depth $L$ and widths $n_1, n_2, ..., n_L$, Jacot et al. (2018) proved that, with GD, the kernel $K_{\theta_t}^L$ converges to a kernel $K^L$ which depends only on $L$ (number of layers) for all $t < T$ when $n_1, n_2, ..., n_L \rightarrow \infty$, where $T$ is an upper bound on the training time, under the technical assumption $\int_0^T ||\nabla_z\ell(f_t(\mathcal{X}, \mathcal{Y}))||_2 dt < \infty$ almost surely with respect to the initialization weights. The infinite width limit of the training dynamics is given by

$$df_t(\mathcal{X}) = -\frac{1}{N}K^L(\mathcal{X}, \mathcal{X})\nabla_z\ell(f_t(\mathcal{X}), \mathcal{Y})dt, \tag{6}$$

We note hereafter $\hat{K}^L = K^L(\mathcal{X}, \mathcal{X})$. As an example, with the quadratic loss $\ell(z, y) = \frac{1}{2}||z - y||^2$, equation 6 is equivalent to

$$df_t(\mathcal{X}) = -\frac{1}{N}\hat{K}^L(f_t(\mathcal{X}) - \mathcal{Y})dt, \tag{7}$$

which is a simple linear model that has a closed-form solution given by

$$f_t(\mathcal{X}) = e^{-\frac{1}{N}\hat{K}^L t}f_0(\mathcal{X}) + (I - e^{-\frac{1}{N}\hat{K}^L t})\mathcal{Y}. \tag{8}$$

For general input $x \in \mathbb{R}^d$, we then have

$$f_t(x) = f_0(x) + K^L(x, \mathcal{X})K^L(\mathcal{X}, \mathcal{X})^{-1}(I - e^{-\frac{1}{N}\hat{K}^L t})(\mathcal{Y} - f_0(\mathcal{X})) \tag{9}$$

Note that in order for $f_t(x)$ to be defined, $\hat{K}^L$ must be invertible. Indeed, it turns out that training with dynamics 6 is only possible if the NTK is invertible. We shed light on this behaviour in the following Lemma.

**Lemma 1** (Trainability of the Neural Network and Invertibility of the NTK). *Assume $f_0(\mathcal{X}) \neq \mathcal{Y}$. Then with dynamics defined by equation 8, $||f_t(\mathcal{X}) - \mathcal{Y}||$ converges to 0 as $t \rightarrow \infty$ if and only if $\hat{K}^L$ is non-singular.*
*Moreover, if $\hat{K}^L$ is singular, there exists a constant $C > 0$ such that for all $t > 0$,*

$$||f_t(\mathcal{X}) - \mathcal{Y}|| \geq C$$

Lemma 1 shows that an invertible NTK is crucial for trainability. Since $K_{\theta_t}^L = K^L$ is constant w.r.t to training time, it is completely determined at the initialization step. It is therefore intuitive to study the impact of the initialization on the NTK, particularly as the number of layers $L$ grows (Deep Neural Networks), which is our focus in this paper. Another interesting aspect is the impact of the NTK on the generalization error of the neural network model. To see this, if the NTK is constant for example (i.e. there exists a constant $\delta$ such as $K^L(x, x') = \delta$ for all $x \neq x'$, this example is useful in the next section), then the second part of $f_t(x)$ in equation 9 is constant w.r.t $x$. Therefore, the generalization function $f_t(x)$ of the model 9 is entirely given by its value at time zero $f_0(x)$, which means that the generalisation error $\mathbb{E}_{x,y}[||f_t(x) - y||]$ remains of order $\mathcal{O}(1)$.

In the next section, we show that the initialization and the activation function have major impact on the invertibility and 'expressivity' of NTK. More precisely, we show that :

1. Under some constraints, the NTK $K^L$ (or a scaled version of the NTK) converges to a limiting NTK $K^\infty$ as $L$ goes to infinity (otherwise it diverges)

2. A special initialization known as the Edge of Chaos (EOC) leads to an invertible $K^\infty$ which makes it useful for training DNNs

3. The EOC initialization gives a sub-exponential rate for this convergence (w.r.t $L$), which means for the same depth $L$, the EOC gives 'richer' limiting NTK, and therefore leading to better generalization properties

4. The smoothness of the activation can further slow this convergence, leading to 'richer' limiting NTK (the convergence to the limiting trivial kernel is slower)

5. Adding Residual connections leads to sub-exponential convergence rate for the NTK (w.r.t to $L$) and we no longer need the Edge of Chaos

# 4 IMPACT OF THE INITIALIZATION AND THE ACTIVATION FUNCTION ON THE NEURAL TANGENT KERNEL

In this section we study the impact of the initialization and the activation function on the limiting NTK for Fully-connected Feed-forward Neural Networks (FFNN). We prove that only an initialization on the Edge of Chaos (EOC) leads to an invertible NTK for deep neural networks. All other initializations will lead to a trivial non-invertible NTK. We also show that the smoothness of the activation function plays a major role in the behaviour of NTK. To simplify notations, we restrict ourselvs to the case $s(x) = x$ and $o = 1$, since generalization to any function $s$ and any $n_L$ is straightforward.

Consider a FFNN of depth $L$, widths $(n_l)_{1 \leq l \leq L}$, weights $w^l$ and bias $b^l$. For some input $x \in \mathbb{R}^d$, the forward propagation is given by

$$y_i^1(x) = \sum_{j=1}^d w_{ij}^1 x_j + b_i^1, \quad y_i^l(x) = \sum_{j=1}^{n_{l-1}} w_{ij}^l \phi(y_j^{l-1}(x)) + b_i^l, \quad \text{for } l \geq 2, \tag{10}$$

where $\phi$ is the activation function.

We initialize the model with $w_{ij}^l \overset{iid}{\sim} \mathcal{N}(0, \frac{\sigma_w^2}{n_{l-1}})$ and $b_i^l \overset{iid}{\sim} \mathcal{N}(0, \sigma_b^2)$, where $\mathcal{N}(\mu, \sigma^2)$ denotes the normal distribution of mean $\mu$ and variance $\sigma^2$. For some $x$, we denote by $q^l(x)$ the variance of $y^l(x)$. The convergence of $q^l(x)$ as $l$ increases is studied in Lee et al. (2018), Schoenholz et al. (2017) and Hayou et al. (2019). In particular, under weak regularity conditions they prove that $q^l(x)$ converges to a point $q(\sigma_b, \sigma_w) > 0$ independent of $x$ as $l \to \infty$. Also the asymptotic behaviour of the correlations between $y^l(x)$ and $y^l(x')$ for any two inputs $x$ and $x'$ is driven by $(\sigma_b, \sigma_w)$; the authors define the EOC as the set of parameters $(\sigma_b, \sigma_w)$ such that $\sigma_w^2 \mathbb{E}[\phi'(\sqrt{q(\sigma_b, \sigma_w)}Z)^2] = 1$ where $Z \sim \mathcal{N}(0, 1)$. Similarly the Ordered, resp. Chaotic, phase is defined by $\sigma_w^2 \mathbb{E}[\phi'(\sqrt{q(\sigma_b, \sigma_w)}Z)^2] < 1$, resp. $\sigma_w^2 \mathbb{E}[\phi'(\sqrt{q(\sigma_b, \sigma_w)}Z)^2] > 1$; more details are recalled in Section 2 of the supplementary material. It turns out that the EOC plays also a crucial role on the NTK. Let us first define two classes of activation functions.

**Definition 1.** *Let $\phi : \mathbb{R} \to \mathbb{R}$ be a measurable function. Then*

1. *$\phi$ is said to be ReLU-like if there exist $\lambda, \beta \in \mathbb{R}$ such that $\phi(x) = \lambda x$ for $x > 0$ and $\phi(x) = \beta x$ for $x \leq 0$.*

2. *$\phi$ is said to be in $\mathcal{S}$ if $\phi(0) = 0$, $\phi$ is twice differentiable, and there exist $n \geq 1$, a partition $(A_i)_{1 \leq i \leq n}$ of $\mathbb{R}$ and infinitely differentiable functions $g_1, g_2, ..., g_n$ such that $\phi^{(2)} = \sum_{i=1}^n 1_{A_i} g_i$, where $\phi^{(2)}$ is the second derivative of $\phi$.*

The class of ReLU-like activations includes ReLU and Leaky-ReLU, whereas the $\mathcal{S}$ class includes, among others, Tanh, ELU and SiLU (Swish). The following proposition establishes that any initialization on the Ordered or Chaotic phase, leads to a trivial limiting NTK as the number of layers $L$ becomes large.

**Proposition 1** (Limiting Neural Tangent Kernel with Ordered/Chaotic Initialization)**.** *Let $(\sigma_b, \sigma_w)$ be either in the ordered or in the chaotic phase. Then, there exist $\lambda, \gamma > 0$ such that*

$$\sup_{x,x' \in \mathbb{R}^d} |K^L(x, x') - \lambda| \leq e^{-\gamma L} \to_{L \to \infty} 0$$

As a result, as $L$ goes to infinity, $K^L$ converges to a constant kernel $K^\infty(x, x') = \lambda$ for all $x, x' \in \mathbb{R}^d$. The training is then impossible. Indeed, we have $K^L(\mathcal{X}, \mathcal{X}) \approx \lambda U$ where $U$ is the matrix with all elements equal to one, i.e. $\hat{K}^L$ is at best degenerate and asymptotically (in $L$) non invertible, rendering the training impossible by Lemma 1. We illustrate empirically this result in Section 5.

Recall that the (matrix) NTK for input data $\mathcal{X}$ is given by

$$K_{\theta_t}^L(\mathcal{X}, \mathcal{X}) = \nabla_\theta f(\mathcal{X}, \theta_t) \nabla_\theta f(\mathcal{X}, \theta_t)^T = \sum_{l=1}^L \nabla_{\theta_l} f(\mathcal{X}, \theta_t) \nabla_{\theta_l} f(\mathcal{X}, \theta_t)^T$$

As shown in Schoenholz et al. (2017) and Hayou et al. (2019), an initialization on the EOC preserves the norm of the gradient as it back-propagates through the network. This means that the terms $\nabla_{\theta_l} f(\mathcal{X}, \theta_t) \nabla_{\theta_l} f(\mathcal{X}, \theta_t)^T$ are of the same order. Hence, it is more convenient to study the average NTK (ANTK hereafter) given by $K^L/L$. Note that the invertibility of the NTK is equivalent to that of the ANTK. The next proposition shows that on the EOC, the ANTK converges to an invertible kernel as $L \to \infty$ at a sub-exponential rate. Moreover, by choosing an activation function in the class $\mathcal{S}$, we can slow the convergence of ANTK with respect to $L$, which means that, for the same depth $L$, a smooth activation function from the class $\mathcal{S}$ leads to 'richer' NTK which is crucial for the generalization error of deep models as discussed in Section 3. This confirms the findings in (Hayou et al., 2019).

**Proposition 2** (Neural Tangent Kernel on the Edge of Chaos). *Let $\phi$ be a non-linear activation function and $(\sigma_b, \sigma_w) \in EOC$.*

1. *If $\phi$ is ReLU-like, then for all $x \in \mathbb{R}^d$, $\frac{K^L(x,x)}{L} = \frac{\sigma_w^2 ||x||^2}{d} + \frac{K^0(x,x)}{L}$. Moreover, there exist $A, \lambda \in (0, 1)$ such that*

$$\sup_{x \neq x' \in \mathbb{R}^d} \left| \frac{K^L(x, x')}{L} - \lambda \frac{\sigma_w^2}{d} ||x|| ||x'||| \right| \leq \frac{A}{L}, \quad K_\infty(x, x') = \frac{\sigma_w^2 ||x|| ||x'||}{d} (1 - (1-\lambda) \mathbf{1}_{x \neq x'})$$

2. *If $\phi$ is in $\mathcal{S}$, then, there exists $q > 0$ such that $\frac{K^L(x,x)}{L} = q + \frac{K^0(x,x)}{L} \to q$. Moreover, there exist $B, C, \lambda \in (0, 1)$ such that*

$$\frac{B \log(L)}{L} \leq \sup_{x \neq x' \in \mathbb{R}^d} \left| \frac{K^L(x, x')}{L} - q\lambda \right| \leq \frac{C \log(L)}{L}, \quad K_\infty(x, x') = q(1 - (1-\lambda) \mathbf{1}_{x \neq x'})$$

Since $0 < \lambda < 1$, on the EOC there exists a matrix $J$ invertible such that $K^L(\mathcal{X}, \mathcal{X}) = L \times J(1 + o(1))$ as $L \to \infty$. Hence, although the NTK grows linearly with $L$, it remains asymptotically invertible. This makes the training possible for deep neural networks when initialized on the EOC, contrariwise to an initialization on the Ordered/Chaotic phase, see Proposition 1). However the limiting kernels $K_\infty$ carry (almost) no information on $x, x'$ and have therefore little expressive power. Interestingly the convergence rate of the ANTK to $K_\infty$ is slow in $L$ ($\mathcal{O}(L^{-1})$ for ReLU-like activation functions and $\mathcal{O}(\log(L)L^{-1})$ for activation functions of type $\mathcal{S}$). This means that as $L$ grows, the NTK remains expressive compared to the Ordered/Chaotic phase case (exponential convergence rate). This is particularly important for the generalization part (see equation 9). The $\log(L)$ gain obtained when using smooth activation functions of type $\mathcal{S}$ means we can train deeper neural networks with this kind of activation functions compared to the ReLU-like activation functions and could explain why ELU and Tanh tend to perform better than ReLU and Leaky-ReLU (see Section 5).

Another important feature of deep neural network which is known to be highly influential is their architecture. The next proposition shows that adding residual connections to a ReLU network leads to a polynomial rate for wide range of initialization parameters.

**Proposition 3** (Residual connections). *Consider the following network architecture (FFNN with residual connections)*

$$y_i^l(x) = y_i^{l-1}(x) + \sum_{j=1}^{n_{l-1}} w_{ij}^l \phi(y_j^{l-1}(x)) + b_i^l, \quad \text{for } l \geq 2. \tag{11}$$

*with initialization parameters $\sigma_b = 0$ and $\sigma_w > 0$ and $\phi$ is the ReLU activation function. Let $K_{res}^L$ be the corresponding NTK. Then for all $x \in \mathbb{R}^d$, $\frac{K_{res}^L(x,x)}{\alpha_L \times 2^L} = \frac{||x||^2}{d} + \mathcal{O}(\gamma_L)$ and there exists $\lambda \in (0, 1)$ such that*

$$\sup_{x \neq x' \in \mathbb{R}^d} \left| \frac{K_{res}^L(x, x')}{\alpha_L \times 2^L} - \frac{||x|| \times ||x'||}{d} \lambda \right| = \mathcal{O}(L^{-1}),$$

*where $\alpha_l$ and $\gamma_l$ are given by*

- *if $\sigma_w < \sqrt{2}$, then $\alpha_L = 1$ and $\gamma_L = (\frac{1 + \sigma_w^2/2}{2})^L$*

- *if $\sigma_w = \sqrt{2}$, then $\alpha_L = L$ and $\gamma_L = L^{-1}$*

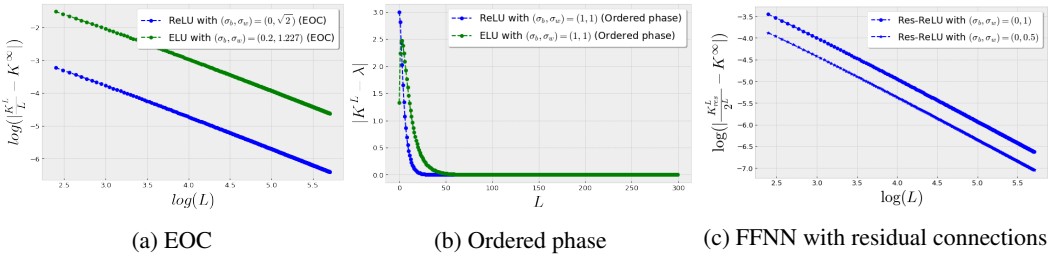

(a) EOC       (b) Ordered phase       (c) FFNN with residual connections

Figure 1: Convergence rates for different initializations and architectures. (a) Edge of Chaos. (b) Ordered phase. (c) Adding residual connections.

- *if $\sigma_w > \sqrt{2}$, then $\alpha_L = (\frac{1+\sigma_w^2/2}{2})^L$ and $\gamma_L = (\frac{1+\sigma_w^2/2}{2})^{-L}$*

Proposition 3 shows that the NTK of a ReLU FFNN with residual connections explodes exponentially with respect to $L$. However, the normalised kernel $K_{res}^L(x, x')/\alpha_L 2^L$ where $x \neq x'$ converges to a limiting kernel similar to $K_\infty$ with a rate $\mathcal{O}(L^{-1})$ for all $\sigma_w > 0$. We say that residual networks 'live' on the Edge of Chaos, i.e. no matter what the choice of $\sigma_w$ is, the convergence rate of the NTK w.r.t $L$ is polynomial and there is no Ordered/Chaotic phase in this case. This could potentially explain why residual networks perform better than FFNN (RELU) in many tasks when the initialization is not on the EOC. We illustrate this result in section 5.

## 5 EXPERIMENTS

In this section, we illustrate empirically the theoretical results obtained in the previous sections. We first illustrate the results of Propositions 1, 2 and 3. Then, we confirm the impact of the EOC and Activation function on the overall performance of the model (FFNN), on MNIST and CIFAR10 datasets.

### 5.1 CONVERGENCE RATE OF $K^L$ AS $L$ GOES TO INFINITY

Propositions 1, 2 and 3 give theoretical convergence rates for quantities of the form $\left| \frac{K^L}{\alpha_L} - K^\infty \right|$. We illustrate these results in Figure 1. Figure 1a shows a convergence rate approximately equal to $\mathcal{O}(L^{-1})$ for ReLU and ELU. Recall that for ELU the exact rate is $\mathcal{O}(\log(L)L^{-1})$ but one cannot observe experimentally the logarithmic factor. However, ELU performs indeed better than ReLU (see Table 1) which might be explained by this $\log(L)$ factor. Figure 1b demonstrates that this convergence occurs at an exponential convergence rate in the Ordered phase for both ReLU and ELU, and Figure 1c the convergence rate in the case of FFNN with residual connections. As predicted by Proposition 3, the convergence rate $\mathcal{O}(L^{-1})$ is independent of the parameter $\sigma_w$.

### 5.2 IMPACT OF THE INITIALIZATION AND SMOOTHNESS OF THE ACTIVATION ON THE OVERALL PERFORMANCE

We train FFNN of width 300 and depths $L \in \{200, 300\}$ and width $\in \{200, 300\}$ with SGD and categorical cross-entropy loss. Training with full batch GD is practically impossible for DNNs, so we use SGD instead (see Section D in the Appendix for more details about how the results extend to SGD) with a batchsize of 64 and a learning rate $10^{-3}$ for $L = 100$ and $10^{-4}$ for $L \in 200, 300$ (this learning rate was found by a grid search of exponential step size 10). For each activation function, we use an initialization on the EOC when it exists, we add the symbol (EOC) after the activation when this is satisfied. We use $(\sigma_b, \sigma_w) = (0, \sqrt{2})$ for ReLU, $(\sigma_b, \sigma_w) = (0.2, 1.227)$ for ELU and $(\sigma_b, \sigma_w) = (0.2, 1.302)$ for Tanh. These values are all on the EOC (see Hayou et al. (2019) for more details). Table 1 displays the test accuracy for different activation functions on MNIST and CIFAR10 after 10 and 100 training epochs for depth 300 and width 300. Functions in class $\mathcal{S}$ (ELU and Tanh) perform much better than ReLU-like activation functions (ReLU, Leaky-Relu$-\alpha$ with $\alpha \in \{0.01, 0.02, 0.03\}$). Even with Parametric ReLU (PReLU) where the parameter of the leaky-ReLU is also learned by backpropagation, we obtain only a small improvement over ReLU. For

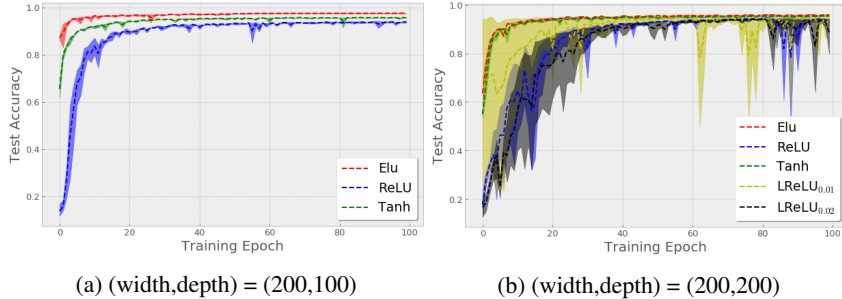

(a) (width,depth) = (200,100)  (b) (width,depth) = (200,200)

Figure 2: Test accuracy for different Activation Functions and (width, depth) on MNIST

Table 1: Test accuracy for a FFNN with width 300 and depth 300 for different activation functions on MNIST and CIFAR10. We show test accuracy after 10 epochs and 100 epochs

|  | MNIST | | CIFAR10 | |
| --- | --- | --- | --- | --- |
| Activation | Epoch 10 | Epoch 100 | Epoch 10 | Epoch 100 |
| ReLU (EOC) | $46.53 \pm 12.01$ | $82.11 \pm 4.51$ | $20.38 \pm 1.85$ | $35.88 \pm 0.6$ |
| $\text{LReLU}_{0.01}$ (EOC) | $48.10 \pm 3.31$ | $84.71 \pm 3.39$ | $22.62 \pm 1.15$ | $29.44 \pm 4.14$ |
| $\text{LReLU}_{0.02}$ (EOC) | $49.09 \pm 3.58$ | $84.3. \pm 3.98$ | $18.62 \pm 4.56$ | $30.78 \pm 6.33$ |
| $\text{LReLU}_{0.03}$ (EOC) | $50.94 \pm 4.48$ | $85.49 \pm 2.71$ | $21.19 \pm 6.53$ | $34.54 \pm 2.32$ |
| PReLU | $51.94 \pm 5.51$ | $87.49 \pm 1.58$ | $22.95 \pm 3.57$ | $36.13 \pm 3.83$ |
| ELU (EOC) | $\mathbf{91.63 \pm 2.21}$ | $\mathbf{96.07 \pm 0.13}$ | $\mathbf{33.81 \pm 1.55}$ | $\mathbf{46.14 \pm 1.49}$ |
| Tanh (EOC) | $91.16 \pm 1.21$ | $95.75 \pm 0.27$ | $32.37 \pm 1.88$ | $42.40 \pm 1.13$ |
| Softplus | $10.11 \pm 0.09$ | $10.13 \pm 0.18$ | $11.13 \pm 0.15$ | $11.09 \pm 0.36$ |
| Sigmoid | $9.85 \pm 0.11$ | $9.87 \pm 0.10$ | $10.65 \pm 0.25$ | $10.33 \pm 0.17$ |

activation functions that do not have an EOC, such as Softplus and Sigmoid, we use He initialization for MNIST and Glorot initialization for CIFAR10 (see He et al. (2015) and Glorot & Bengio (2010)). For Softplus and Sigmoid, the training algorithm is stuck at a low test accuracy $\sim 10\%$ which is the test accuracy of a uniform random classifier with 10 classes.

## 6 CONCLUSION AND LIMITATIONS

That the training dynamics of deep neural networks is equivalent to a Functional Gradient Descent with respect to the Neural Tangent Kernel. In the infinite width limit, the NTK has a closed-form expression. This approximation sheds light on how the NTK impacts the training dynamics: it controls the training rate and the generalization function. Using this approximation for wide neural networks (Mean-field approximation), we show that for an initialization in the Ordered/Chaotic phase, NTK converges exponentially fast to a non-invertible kernel as the number of layers goes to infinity, making training impossible. An initialization on the EOC leads to an invertible ANTK (and NTK) even for an infinite number of layers: the convergence rate is $\mathcal{O}(L^{-1})$ for ReLU-like activation functions and $\mathcal{O}(\log(L)L^{-1})$ for a class of smooth activation functions.

However, recent findings showed that the infinite width approximation of the NTK does not fully capture the dynamics of the training of DNNs. A recent line of work showed that the NTK for finite width neural networks changes with time and might even be random (Chizat & Bach (2018), Ghorbani et al. (2019), Huang & Yau (2019), Arora et al. (2019)). Therefore, we believe that the NTK is a useful tool to partially understand wide deep neural networks (have insights on hyper-parameters choices for example) and not a tool to train neural networks.

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

We provide in Section A and Section B the proof of the theoretical results presented in the main document. Section C provides additional theoretical results while Section **??** presents additional experimental results.

# A    APPENDIX: PROOFS OF SECTION 3: NEURAL NETWORKS AND NEURAL TANGENT KERNEL

**Lemma 1** (Trainability of the Neural Network and Invertibility of the NTK). *Assume $f_0(\mathcal{X}) \neq \mathcal{Y}$. Then with dynamics (8), $||f_t(\mathcal{X}) - \mathcal{Y}||$ converges to 0 as $t \to \infty$ if and only if $\hat{K}^L$ is non-singular. Moreover, if $K^L$ is singular, there exists a constant $C > 0$ such that for all $t > 0$,*

$$||f_t(\mathcal{X}) - \mathcal{Y}|| \geq C$$

*Proof.* Assume $f_0(\mathcal{X}) \neq \mathcal{Y}$. Let $\hat{K}^L = Q^T D Q$ be the spectral decomposition of the empirical NTK; i.e. $Q$ is an orthogonal matrix and $D$ is a diagonal matrix.

We have that $e^{-\frac{1}{N}\hat{K}^L t} = Q^T e^{-\frac{1}{N}Dt} Q = Q^T Diag(e^{-\frac{d_i}{N}t})_{1 \leq i \leq oN} Q$ where $(d_i)_{1 \leq i \leq oN}$ are the eigenvalues. We also have $||f_t(\mathcal{X}) - \mathcal{Y}|| = ||e^{-\frac{1}{N}\hat{K}^L t}(f_0(\mathcal{X}) - \mathcal{Y})||$. Therefore, the equivalence holds true.

Moreover, assume $\hat{K}^L$ in singular. Let $Z_t = Q(f_t(\mathcal{X}) - \mathcal{Y})Q^T$. We have that $Z_t = e^{-\frac{1}{N}Dt} Z_0$. Since $D$ has at least one zero diagonal value, then there exists $j \in \{1, 2, ..., oN\}$ such that for all $t$, $(Z_t)_j = (Z_0)_j$, and we have

$$\begin{aligned} ||f_t(\mathcal{X}) - \mathcal{Y}|| &= ||Z_t|| \\ &\geq |(Z_t)j| = |(Z_0)j| \end{aligned}$$

$\square$

**Lemma 2** (Discretization Error for Full-Batch Gradient Descent). *Assume $\nabla_\theta \mathcal{L}$ is $C$-lipschitz, then there exists $C' > 0$ that depends only on $C$ and $T$ such that*

$$\sup_{k \in [0, T/\eta]} ||\theta_{t_k} - \hat{\theta}_k|| \leq \eta C'$$

*Proof.* For $t \in [0, T]$, we define the stepwise constant system $\tilde{\theta}_t = \hat{\theta}_{\lfloor t/\eta \rfloor}$. Let $t \in [0, T]$, we have

$$\begin{aligned} \tilde{\theta}_t &= \theta_0 - \eta \sum_{k=0}^{\lfloor t/\eta \rfloor - 1} \nabla_\theta \mathcal{L}(\hat{\theta}_k) \\ &= \theta_0 - \int_0^t \nabla_\theta \mathcal{L}(\tilde{\theta}_s)ds + \eta \int_{\lfloor * \rfloor t/\eta - 1}^{t/\eta} \nabla_\theta \mathcal{L}(\hat{\theta}_{\lfloor * \rfloor s})ds \end{aligned}$$

Therefore,

$$\begin{aligned} ||\theta_t - \tilde{\theta}_t|| &\leq \int_0^t ||\nabla_\theta \mathcal{L}(\theta_s) - \nabla_\theta \mathcal{L}(\tilde{\theta}_s)||ds + \eta \int_{\lfloor * \rfloor t/\eta - 1}^{t/\eta} ||\nabla_\theta \mathcal{L}(\hat{\theta}_{\lfloor * \rfloor s})||ds \\ &\leq C \int_0^t ||\theta_s - \tilde{\theta}_s||ds + \eta(t/\eta - \lfloor * \rfloor t/\eta)||\nabla_\theta \mathcal{L}(\hat{\theta}_{\lfloor * \rfloor t/\eta})|| + \eta||\nabla_\theta \mathcal{L}(\hat{\theta}_{\lfloor * \rfloor t/\eta - 1})|| \end{aligned}$$

Moreover, for any $k \in [0, \lfloor * \rfloor T/\eta]$, we have

$$\begin{aligned} ||\hat{\theta}_k - \theta_0|| &\leq (1 + \eta C)||\hat{\theta}_{k-1} - \theta_0|| \\ &\leq (1 + \eta C)^{T/\eta}||\hat{\theta}_1 - \theta_0|| \\ &\leq e^{CT}||\hat{\theta}_1 - \theta_0|| \end{aligned}$$

where we have used $\log(1 + \eta C) \leq \eta C$. Using this result, there exists a constant $\tilde{C}$ depending on $T$ and $C$ such that

$$\eta(t/\eta - \lfloor * \rfloor t/\eta)||\nabla_\theta \mathcal{L}(\hat{\theta}_{\lfloor * \rfloor t/\eta})|| + \eta||\nabla_\theta \mathcal{L}(\hat{\theta}_{\lfloor * \rfloor t/\eta - 1})|| \leq \eta(2||\nabla_\theta \mathcal{L}(\hat{\theta}_0)|| + C||\hat{\theta}_{\lfloor * \rfloor t/\eta} - \theta_0|| + C||\hat{\theta}_{\lfloor * \rfloor t/\eta - 1} - \theta_0||)$$
$$\leq \eta\tilde{C}$$

Now we have

$$||\theta_t - \tilde{\theta}_t|| \leq C \int_0^t ||\theta_s - \tilde{\theta}_s||ds + \eta\tilde{C},$$

so we can conclude using Gronwall's lemma. $\qquad\square$

# B  PROOFS OF SECTION 4: IMPACT OF THE INITIALIZATION AND THE ACTIVATION FUNCTION ON THE NEURAL TANGENT KERNEL

We first recall the results obtained in Lee et al. (2018), Schoenholz et al. (2017) and Hayou et al. (2019) where the impact of the EOC (Edge of Chaos) on the initialization is studied. We also present some results that we will be used below.

Consider a FFNN of depth $L$, widths $(n_l)_{1 \leq l \leq L}$, weights $w^l$ and bias $b^l$. For some input $x \in \mathbb{R}^d$, the forward propagation is given by

$$y_i^1(x) = \sum_{j=1}^d w_{ij}^1 x_j + b_i^1, \quad y_i^l(x) = \sum_{j=1}^{n_{l-1}} w_{ij}^l \phi(y_j^{l-1}(x)) + b_i^l, \quad \text{for } l \geq 2, \tag{12}$$

where $\phi$ is the activation function.

We initialize the model with $w_{ij}^l \overset{iid}{\sim} \mathcal{N}(0, \frac{\sigma_w^2}{n_{l-1}})$ and $b_i^l \overset{iid}{\sim} \mathcal{N}(0, \sigma_b^2)$, where $\mathcal{N}(\mu, \sigma^2)$ denotes the normal distribution of mean $\mu$ and variance $\sigma^2$. For some $x$, we denote by $q^l(x)$ the variance of $y^l(x)$. In general, $q^l(x)$ converges to a point $q(\sigma_b, \sigma_w) > 0$ independent of $x$ as $l \to \infty$. The EOC is defined by the set of parameters $(\sigma_b, \sigma_w)$ such that $\sigma_w^2 \mathbb{E}[\phi'(\sqrt{q(\sigma_b, \sigma_w)}Z)^2] = 1$ where $Z \sim \mathcal{N}(0, 1)$. Similarly the Ordered, resp. Chaotic, phase is defined by $\sigma_w^2 \mathbb{E}[\phi'(\sqrt{q(\sigma_b, \sigma_w)}Z)^2] < 1$, resp. $\sigma_w^2 \mathbb{E}[\phi'(\sqrt{q(\sigma_b, \sigma_w)}Z)^2] > 1$ (see Hayou et al. (2019) for more details). For two inputs $x, x' \in \mathbb{R}^d$, define $\Sigma^l(x, x') = \mathbb{E}[y^l(x)y^l(x')]$ and let $c^l(x, x')$ be the corresponding correlation. Let $f$ be the correlation function defined implicitly by $c^{l+1} = f(c^l)$. In the limit of infinitely wide networks, we have the following results (Hayou et al. (2019)) :

- $\Sigma^l(x, x') = \sigma_b^2 + \sigma_w^2 \mathbb{E}_{z \sim \mathcal{N}(0, \Sigma^{l-1})}[\phi(z(x))\phi(z(x'))]$.
- There exist $q, \lambda > 0$ such that, for all $\sup_{x \in \mathbb{R}^d} |\Sigma^l(x, x) - q| \leq e^{-\lambda l}$.
- On the Ordered phase, there exists $\gamma > 0$ such that $\sup_{x, x' \in \mathbb{R}^d} |c^l(x, x') - 1| \leq e^{-\gamma l}$.
- On the chaotic phase, there exist $\gamma > 0$ and $c < 1$ such that $\sup_{x \neq x' \in \mathbb{R}^d} |c^l(x, x') - c| \leq e^{-\gamma l}$.
- For ReLU network on the EOC, we have that $\Sigma^l(x, x) = \frac{\sigma_w^2}{d}||x||^2$ for all $l \geq 1$. Moreover, we have

$$f(x) \underset{x \to 1^-}{=} x + \frac{2\sqrt{2}}{3\pi}(1 - x)^{3/2} + O((1 - x)^{5/2})$$

- In general, we have

$$f(x) = \frac{\sigma_b^2 + \sigma_w^2 \mathbb{E}[\phi(\sqrt{q}Z_1)\phi(\sqrt{q}Z(x))]}{q}$$

where $Z(x) = xZ_1 + \sqrt{1 - x^2}Z_2$ and $Z_1, Z_2$ are iid standard Gaussian variables.

- On the EOC, we have $f'(1) = 1$
- If $\phi$ is $k$-times differentiable, then $f$ is $k$-times differentiable and for all $1 \leq j \leq k$, we have $f^{(j)}(x) = \sigma_w^2 q^{j-1} \mathbb{E}[\phi^{(j)}(Z_1)\phi^{(j)}(Z(x))]$

- From Jacot et al. (2018), we have that

$$K^l(x, x') = K^{l-1}(x, x')\dot{\Sigma}^l(x, x') + \Sigma^l(x, x').$$

where the definition of $\dot{\Sigma}^l(x, x')$ is given in Proposition 1 below.

**Definition 1.** *Let $\phi : \mathbb{R} \to \mathbb{R}$ be a measurable function. Then*

1. *$\phi$ is said to be ReLU-like if there exist $\lambda, \beta \in \mathbb{R}$ such that $\phi(x) = \lambda x$ for $x > 0$ and $\phi(x) = \beta x$ for $x \leq 0$.*

2. *$\phi$ is said to be in $\mathcal{S}$ if $\phi(0) = 0$, $\phi$ is twice differentiable, and there exist $n \geq 1$, a partition $(A_i)_{1 \leq i \leq n}$ of $\mathbb{R}$ and infinitely differentiable functions $g_1, g_2, ..., g_n$ such that $\phi^{(2)} = \sum_{i=1}^n 1_{A_i} g_i$, where $\phi^{(2)}$ is the second derivative of $\phi$.*

The following two lemmas will be useful to prove the results of Section 3 in the main paper.

**Lemma 2.** *Let $(a_l)$ be a sequence of non-negative real numbers such that $\forall l \geq 0, a_{l+1} \leq \alpha a_l + k e^{-\beta l}$, where $\alpha \in (0, 1)$ and $k, \beta > 0$. Then there exists $\gamma > 0$ such that $\forall l \geq 0, l \leq e^{-\gamma l}$.*

*Proof.* Using the inequality on $a_l$, we can easily see that

$$a_l \leq a_0 \alpha^l + k \sum_{j=0}^{l-1} \alpha^j e^{-\beta(l-j)}$$

$$\leq a_0 \alpha^l + k \frac{l}{2} e^{-\beta l/2} + k \frac{l}{2} \alpha^{l/2}$$

where we divided the sum into two parts separated by index $l/2$ and upper-bounded each part. The existence of $\gamma$ is straightforward. □

**Proposition 1** (Limiting Neural Tangent Kernel with Ordered/Chaotic Initialization). *Let $(\sigma_b, \sigma_w)$ be in the ordered or chaotic phase. Then, there exist $\lambda, \gamma > 0$ such that*

$$\sup_{x,x' \in \mathbb{R}^d} |K^L(x, x') - \lambda| \leq e^{-\gamma L} \to_{L \to \infty} 0$$

*Proof.* From Jacot et al. (2018), we have that

$$K^l(x, x') = K^{l-1}(x, x')\dot{\Sigma}^l(x, x') + \Sigma^l(x, x')$$

where $\Sigma^1(x, x') = \sigma_b^2 + \frac{\sigma_w^2}{d} x^T x'$ and $\Sigma^l(x, x') = \sigma_b^2 + \sigma_w^2 \mathbb{E}_{f \sim \mathcal{N}(0, \Sigma^{l-1})}[\phi(f(x))\phi(f(x'))]$ and $\dot{\Sigma}^l(x, x') = \mathbb{E}_{f \sim \mathcal{N}(0, \Sigma^{l-1})}[\phi'(f(x))\phi'(f(x'))]$. In the ordered/chaotic phase, Hayou et al. (2019) showed that there exist $k, \gamma, l_0 > 0$ and $\alpha \in (0, 1)$ such that for all $l \geq l_0$ we have

$$\sup_{x,x' \in \mathbb{R}^d} |\Sigma^l(x, x') - k| \leq e^{-\gamma l}$$

and

$$\sup_{x,x' \in \mathbb{R}^d} \dot{\Sigma}^l(x, x') \leq \alpha.$$

Therefore we have for any $l \geq l_0$ and $x, x' \in \mathbb{R}^d$

$$K^l(x, x') \leq \alpha K^{l-1}(x, x') + k + e^{-\gamma l}.$$

Letting $r_l = K^l(x, x') - \frac{k}{1-\alpha}$, we have

$$r_l \leq \alpha r_{l-1}.$$

We can now conclude using Lemma 2. □

Now, we show that the Initialization on the EOC leads to an invertible NTK even if the number of layers $L$ goes to infinity. We first prove two preliminary lemmas that will be useful for the proof of the next proposition.

**Lemma 3.** *Let* $(a_l), (b_l), (\lambda_l)$ *be three sequences of real numbers such that*

$$a_l = a_{l-1}\lambda_l + b_l$$

$$\lambda_l = 1 - \frac{\alpha}{l} + O(l^{-1-\beta})$$

$$b_l = q + O(l^{-1})$$

*where* $\alpha \in \mathbb{N}^*, \beta, q \in \mathbb{R}^+$ *and* $\alpha > \beta - 1$.
*Then,*

$$\frac{a_l}{l} = \frac{q}{1+\alpha} + O(l^{-\min(1,\beta)})$$

*Proof.* It is easy to see that there exists a constant $G > 0$ $|a_l| \le G \times l + |a_0|$ for all $l \ge 0$, therefore $(a_l/l)$ is bounded. Now let $\zeta = \min(1, \beta)$ and $r_l = \frac{a_l}{l}$. We have

$$r_l = r_{l-1}(1 - \frac{1}{l})(1 - \frac{\alpha}{l} + O(l^{-1-\beta})) + \frac{q}{l} + O(l^{-2})$$

$$= r_{l-1}(1 - \frac{1+\alpha}{l}) + \frac{q}{l} + O(l^{-1-\zeta}).$$

Letting $x_l = \left| r_l - \frac{q}{1+\alpha} \right|$, there is exist a constant $M > 0$ such that

$$x_l \le x_{l-1}(1 - \frac{1+\alpha}{l}) + \frac{M}{l^{1+\zeta}}.$$

Hence, we have

$$x_l \le x_0 \prod_{k=1}^{l}(1 - \frac{1+\alpha}{k}) + M \sum_{k=1}^{l} \prod_{j=k+1}^{l}(1 - \frac{1+\alpha}{j})\frac{1}{k^{1+\zeta}}.$$

By taking the logarithm of the first term in the right hand side and using the fact that $\sum_{k=1}^{l}\frac{1}{k} \sim \log(l)$, we have

$$\prod_{k=1}^{l}(1 - \frac{1+\alpha}{k}) \sim l^{-1-\alpha}$$

For the second part, observe that

$$\prod_{j=k+1}^{l}(1 - \frac{1+\alpha}{j}) = \frac{(l-\alpha-1)!}{l!}\frac{k!}{(k-\alpha-1)!}$$

and

$$\frac{k!}{(k-\alpha-1)!}\frac{1}{k^{1+\zeta}} \sim_{k\to\infty} k^{\alpha-\zeta}$$

so that,

$$\sum_{k=1}^{l}\frac{k!}{(k-\alpha-1)!}\frac{1}{k^{1+\zeta}} \sim \sum_{k=1}^{l} k^{\alpha-\zeta}$$

$$\sim \int_{1}^{l} t^{\alpha-\zeta}dt$$

$$\sim \frac{1}{\alpha-\zeta+1}l^{\alpha-\zeta+1}$$

therefore,

$$\sum_{k=1}^{l} \prod_{j=k+1}^{l}(1 - \frac{1+\alpha}{j})\frac{1}{k^{1+\zeta}} = \frac{(l-\alpha-1)!}{l!}\sum_{k=1}^{l}\frac{k!}{(k-\alpha-1)!}\frac{1}{k^{1+\zeta}}$$

$$\sim \frac{1}{\alpha-\zeta+1}l^{-\zeta}$$

We can now conclude using the fact that $\alpha > \beta - 1$.

$\square$

We now introduce a different form of the previous Lemma that will be useful for other applications.

**Lemma 4.** *Let $(a_l), (b_l), (\lambda_l)$ be three sequences of real numbers such that*

$$a_l = a_{l-1}\lambda_l + b_l$$

$$\lambda_l = 1 - \frac{\alpha}{l} + \kappa\frac{\log(l)}{l^2} + O(l^{-1-\beta})$$

$$b_l = q + O(l^{-1})$$

*where $\alpha \in \mathbb{N}^*, \beta, q \in \mathbb{R}^+$ and $\alpha > \beta - 1, \beta \geq 1$.*
*Then, there exists $A, B > 0$ such that*

$$A\frac{\log(l)}{l} \leq |\frac{a_l}{l} - \frac{q}{1+\alpha}| \leq B\frac{\log(l)}{l}$$

*Proof.* It is easy to see that there exists a constant $G > 0$ $|a_l| \leq G \times l + |a_0|$ for all $l \geq 0$, therefore $(a_l/l)$ is bounded. Let $r_l = \frac{a_l}{l}$. We have

$$r_l = r_{l-1}(1 - \frac{1}{l})(1 - \frac{\alpha}{l} + \kappa\frac{\log(l)}{l^2} + O(l^{-1-\beta})) + \frac{q}{l} + O(l^{-2})$$

$$= r_{l-1}(1 - \frac{1+\alpha}{l}) + r_{l-1}\kappa\frac{\log(l)}{l^2} + \frac{q}{l} + O(l^{-2})$$

Let $x_l = r_l - \frac{q}{1+\alpha}$. It is clear that $\lambda_l = 1 - \alpha/l + O(l^{-3/2})$. Therefore, using Lemma 3 with $\beta = 1/2$, we have $r_l \to \frac{q}{1+\alpha}$. Thus, there exists $\kappa_1, \kappa_2, M, l_0 > 0$ such that for all $l \geq l_0$

$$x_{l-1}(1 - \frac{1+\alpha}{l}) + \kappa_1\frac{\log(l)}{l^2} - \frac{M}{l^2} \leq x_l \leq x_{l-1}(1 - \frac{1+\alpha}{l}) + \kappa_2\frac{\log(l)}{l^2} + \frac{M}{l^2}$$

Similarly to the proof of Lemma 3, it follows that

$$x_l \leq x_{l_0}\prod_{k=l_0}^{l}(1 - \frac{1+\alpha}{k}) + \sum_{k=l_0}^{l}\prod_{j=k+1}^{l}(1 - \frac{1+\alpha}{j})\frac{\kappa_2\log(k) + M}{k^2}$$

and

$$x_l \geq x_0\prod_{k=0}^{l}(1 - \frac{1+\alpha}{k}) + \sum_{k=l_0}^{l}\prod_{j=k+1}^{l}(1 - \frac{1+\alpha}{j})\frac{\kappa_1\log(k) - M}{k^2}$$

Recall that we have

$$\prod_{k=1}^{l}(1 - \frac{1+\alpha}{k}) \sim l^{-1-\alpha}$$

and

$$\prod_{j=k+1}^{l}(1 - \frac{1+\alpha}{j}) = \frac{(l-\alpha-1)!}{l!}\frac{k!}{(k-\alpha-1)!}$$

so that

$$\frac{k!}{(k-\alpha-1)!}\frac{\kappa_1\log(k) - M}{k^2} \sim_{k\to\infty} \log(k)k^{\alpha-1}$$

Therefore, we obtain

$$\sum_{k=1}^{l}\frac{k!}{(k-\alpha-1)!}\frac{\kappa_1\log(k) - M}{k^2} \sim \sum_{k=1}^{l}\log(k)k^{\alpha-1}$$

$$\sim \int_{1}^{l}\log(t)t^{\alpha-1}dt$$

$$\sim C_1 l^\alpha \log(\alpha)$$

where $C_1 > 0$ is a constant. Similarly, there exists a constant $C_2 > 0$ such that

$$\sum_{k=1}^{l} \frac{k!}{(k-\alpha-1)!} \frac{\kappa_2 \log(k) + M}{k^2} \sim C_2 l^\alpha \log(\alpha)$$

We conclude using the fact that $\frac{(l-\alpha-1)!}{l!} \sim l^{-1-\alpha}$.

$\square$

**Proposition 2** (Neural Tangent Kernel on the Edge of Chaos). *Let $\phi$ be a non-linear activation function and $(\sigma_b, \sigma_w) \in EOC$.*

1. *If $\phi$ is ReLU-like, then for all $x \in \mathbb{R}^d$, $\frac{K^L(x,x)}{L} = \frac{\sigma_w^2 ||x||^2}{d} + \frac{K^0(x,x)}{L}$. Moreover, there exist $A, \lambda \in (0,1)$ such that*

$$\sup_{x \neq x' \in \mathbb{R}^d} \left| \frac{K^L(x,x')}{L} - \lambda \frac{\sigma_w^2}{d} ||x|| ||x'|| \right| \leq \frac{A}{L}$$

2. *If $\phi$ is in $\mathcal{S}$, then, there exist $q > 0$ such that $\frac{K^L(x,x)}{L} = q + \frac{K^0(x,x)}{L} \to q$. Moreover, there exist $B, C, \lambda \in (0,1)$ such that*

$$\frac{B \log(L)}{L} \leq \sup_{x \neq x' \in \mathbb{R}^d} \left| \frac{K^L(x,x')}{L} - q\lambda \right| \leq \frac{C \log(L)}{L}$$

*Proof.* We use some results from Hayou et al. (2019) in this proof.

Let $x, x' \in \mathbb{R}^d$ and $c_{x,x'}^l = \frac{\Sigma(x,x')}{\sqrt{\Sigma(x,x)\Sigma(x',x')}}$. Let $\gamma_l := 1 - c_{x,x'}^l$ and $f$ be the correlation function defined by the recursive equation $c^{l+1} = f(x^l)$. From the preliminary results, we know that $\Sigma^l(x,x) = \frac{\sigma_w^2}{d} ||x||^2$ and that $K^l(x,x') = K^{l-1}(x,x')\dot{\Sigma}^l(x,x') + \Sigma^l(x,x')$. This concludes the proof for $K^L(x,x)$. We denote $s = \frac{2\sqrt{2}}{3\pi}$. From Hayou et al. (2019), we have on the EOC $\gamma_{l+1} = \gamma_l - s\gamma_l^{3/2} + O(\gamma_l^{5/2})$ so that

$$\gamma_{l+1}^{-1/2} = \gamma_l^{-1/2}(1 - s\gamma_l^{1/2} + O(\gamma_l^{3/2}))^{-1/2} = \gamma_l^{-1/2}(1 + \frac{s}{2}\gamma_l^{1/2} + O(\gamma_l^{3/2}))$$

$$= \gamma_l^{-1/2} + \frac{s}{2} + O(\gamma_l).$$

Thus, as $l$ goes to infinity

$$\gamma_{l+1}^{-1/2} - \gamma_l^{-1/2} \sim \frac{s}{2}$$

and by summing and equivalence of positive divergent series

$$\gamma_l^{-1/2} \sim \frac{s}{2} l.$$

Moreover, since $\gamma_{l+1}^{-1/2} - \gamma_l^{-1/2} = \frac{s}{2} + O(\gamma_l) = \frac{s}{2} + O(l^{-2})$, we have $\gamma_l^{-1/2} = \frac{s}{2} l + O(1)$. Therefore, $c_{x,x'}^l = 1 - \frac{9\pi^2}{2l^2} + O(l^{-3})$.
we also have

$$f'(x) = \frac{1}{\pi} \arcsin(x) + \frac{1}{2}$$

$$= 1 - \frac{\sqrt{2}}{\pi}(1-x)^{1/2} + O((1-x)^{5/2}).$$

Thus, it follows that

$$f'(c_{x,x'}^l) = 1 - \frac{3}{l} + O(l^{-2})$$

Moreover, $q_{x,x'}^l = q + O(l^{-2})$ where $q$ is the limiting variance of $y^l$.

Using Lemma 3, we conclude that $\frac{K^l(x,x')}{l} = \frac{1}{4}\frac{\sigma_w^2}{d}||x||||x'|| + O(l^{-1})$. Since $c^{x,x'}$ is bounded, this result is uniform in $x, x'$. Therefore, we can take the supremum over $x, x' \in \mathbb{R}^d$.

2. We prove the result when $\phi^{(2)}(x) = 1_{x<0}g_1(x) + 1_{x\geq0}g_2(x)$. The generalization to the whole class is straightforward. Let $f$ be the correlation function. We first show that for all $k \geq 3$ $f^{(k)}(x) = \frac{1}{(1-x^2)^{(k-2)/2}}g_k(x)$ where $g_k \in \mathcal{C}^\infty$.

We have

$$f''(x) = \sigma_w^2 q\mathbb{E}[\phi''(\sqrt{q}Z_1)\phi''(\sqrt{q}U_2(x))]$$
$$= \sigma_w^2 q\mathbb{E}[\phi''(\sqrt{q}Z_1)1_{U_2(x)<0}g_1(\sqrt{q}U_2(x))] + \sigma_w^2 q\mathbb{E}[\phi''(\sqrt{q}Z_1)1_{U_2(x)>0}g_2(\sqrt{q}U_2(x))].$$

Let $G(x) = \mathbb{E}[\phi''(\sqrt{q}Z_1)1_{U_2(x)<0}g_1(\sqrt{q}U_2(x))]$ then

$$G'(x) = \mathbb{E}[\phi''(\sqrt{q}Z_1)(Z_1 - \frac{x}{\sqrt{1-x^2}}Z_2)\delta_{U_2(x)=0}\frac{1}{\sqrt{1-x^2}}g_1(\sqrt{q}U_2(x))]$$
$$+ \mathbb{E}[\phi''(\sqrt{q}Z_1)1_{U_2(x)<0}\sqrt{q}(Z_1 - \frac{x}{\sqrt{1-x^2}}Z_2)g_1'(\sqrt{q}U_2(x))].$$

It is easy to see that $G'(x) = \frac{1}{\sqrt{1-x^2}}G_1(x)$ where $G_1 \in \mathcal{C}^1$. A similar analysis can be applied to the second term of $f''$. We conclude that there exists $g_3 \in \mathcal{C}^\infty$ such that $f^{(3)}(x) = \frac{1}{\sqrt{1-x^2}}g(x)$. We obtain the result by induction.

Since $f^{(k)}$ are potentially not defined at 1, we use the change of variable $x = 1 - t^2$ to obtain a Taylor expansion near 1. Simple algebra shows that the function $t \to f(1-t^2)$ has a Taylor expansion near 0:

$$f(1-t^2) = 1 - t^2 f'(1) + \frac{t^4}{2}f''(1) + \frac{t^6}{6}f^{(3)}(1) + O(t^8).$$

Therefore,

$$f(x) = 1 + (x-1)f'(1) + \frac{(x-1)^2}{2}f''(1) + \frac{(1-x)^3}{6}f^{(3)}(1) + O((x-1)^4).$$

Letting $\lambda_l := 1 - c^l$, there exist $\alpha, \beta > 0$ such that

$$\lambda_{l+1} = \lambda_l - \alpha\lambda_l^2 - \beta\lambda_l^3 + O(\lambda_l^4)$$

therefore,

$$\lambda_{l+1}^{-1} = \lambda_l^{-1}(1 - \alpha\lambda_l - \beta\lambda^2 + O(\lambda_l^3))^{-1}$$
$$= \lambda_l^{-1}(1 + \alpha\lambda_l + \beta\lambda_l^2 + O(\lambda_l^3))$$
$$= \lambda_l^{-1} + \alpha + \beta\lambda_l + O(\lambda_l^2).$$

By summing (divergent series), we have that $\lambda_l^{-1} \sim \frac{l}{\beta_q}$. Therefore,

$$\lambda_{l+1}^{-1} - \lambda_l^{-1} - \alpha = \beta\alpha^{-1}l^{-1} + O(l^{-2})$$

By summing a second time, we obtain

$$\lambda_l^{-1} = \alpha l + \beta\alpha^{-1}\log(l) + O(1)$$

so that $\lambda_l = \alpha^{-1}l^{-1} - \alpha^{-1}\beta\frac{\log(l)}{l^2} + O(l^{-2})$.

Using the fact that $f'(x) = 1 + (x-1)f''(1) + O((x-1)^2)$, we have $f'(c_{x,x'}^l) = 1 - \frac{2}{l} + \kappa\frac{\log(l)}{l^2} + O(l^{-2})$. We can now conclude using Lemma 4. Using again the agrument of the boundedness of $c_{x,x'}^1$, we can take the supremum.

$\square$

**Proposition 3.** *Consider the following network architecture (FFNN with residual connections)*

$$y_i^l(x) = y_i^{l-1}(x) + \sum_{j=1}^{n_{l-1}} w_{ij}^l \phi(y_j^{l-1}(x)) + b_i^l, \quad for \ l \geq 2. \tag{13}$$

*with initialization parameters $\sigma_b = 0$ and $\sigma_w > 0$. Let $K_{res}^L$ be the corresponding NTK. For all $x \in \mathbb{R}^d$, $\frac{K_{res}^L(x,x)}{L \times 2^L} = \sigma_w^2 \frac{||x||^2}{d} + \mathcal{O}(L^{-1})$ and there exists $\lambda \in (0,1)$ such that*

$$\sup_{x \neq x' \in \mathbb{R}^d} \Big| \frac{K_{res}^L(x,x')}{L \times 2^L} - \sigma_w^2 \frac{||x|| \times ||x'||}{d} \lambda \Big| = \mathcal{O}(L^{-1})$$

.

*Proof.* We only give a sketch of the proof. A more rigorous proof can be easily done but it unecessary. As in the feedforward without residual connections case, it is easy to see that $K_{res}^L$ satisfies the following recursive equation

$$K_{res}^l(x,x') = K_{res}^{l-1}(x,x')(\dot{\Sigma}^l(x,x') + 1) + \Sigma^l(x,x')$$

To see this, with FeedForward Neural Network, we have ( Jacot et al. (2018))

$$\frac{\partial y_i^{l+1}}{\partial \theta_{1:l}}(x) \Big( \frac{\partial y_i^{l+1}}{\partial \theta_{1:l}}(x') \Big)^t = \sum_{j,j'}^{n_l} w_{ij}^{l+1} w_{ij'}^{l+1} \phi'(y_j^l) \frac{\partial y_i^l}{\partial \theta_{1:l}}(x) \Big( \frac{\partial y_i^l}{\partial \theta_{1:l}}(x) \Big)^t + I_W$$

whereas for Residual Networks, we have an extra term

$$\frac{\partial y_i^{l+1}}{\partial \theta_{1:l}}(x) \Big( \frac{\partial y_i^{l+1}}{\partial \theta_{1:l}}(x') \Big)^t = \frac{\partial y_i^l}{\partial \theta_{1:l}}(x) \Big( \frac{\partial y_i^l}{\partial \theta_{1:l}}(x') \Big)^t + \sum_{j,j'}^{n_l} w_{ij}^{l+1} w_{ij'}^{l+1} \phi'(y_j^l) \frac{\partial y_i^l}{\partial \theta_{1:l}}(x) \Big( \frac{\partial y_i^l}{\partial \theta_{1:l}}(x) \Big)^t + I_W'$$

where $I_W, I_W'$ are quantities that converge to 0 as $n_l$ grows (self-averaging). The additional term results in an added term $K^l(x,x')$ in the formula of $K^{l+1}(x,x')$.

we already now that $\dot{\Sigma}^l(x,x) = 1$. Moreover, we have $\Sigma^l(x,x) = \Sigma^{l-1}(x,x) + \sigma_w^2/2\Sigma^{l-1}(x,x) = (1 + \sigma_w^2/2)^{l-1} \frac{\sigma_w^2}{d} ||d||$. Depending on the value of $\sigma_w$, the behaviour of $K^l(x,x')$ changes. However, from Hayou et al. (2019), we have that $\dot{\Sigma}^l(x,x') = 1 - \beta l^{-1} + O(l^{-2})$. So by scaling with $\alpha_L 2^L$ and using Lemma 3, we conclude on the convergence rate of $O(l^{-1})$. We can take the supremum as the result of the boundedness of $c^1(x,x')$.

$\square$

## C   IMPACT OF THE INITIALIZATION THE OUTPUT FUNCTION

In this section, we show how an initialization on the EOC impacts on the output function of the neural network. More precisely, we show that it leads to a larger range compared to an initialization in the Ordered/chaotic phase.

Let us start by a simple Lemma that compares the expectations of some smooth mapping with respect to two different Gaussian vectors.

**Lemma 5.** *Let $X = (X_i)_{1 \leq i \leq n}$, $Y = (Y_i)_{1 \leq i \leq N}$ be two centered Gaussian vectors in $\mathbb{R}^n$. Let $g \in \mathcal{D}^2(\mathbb{R}^n, \mathbb{R})$. Then we have*

$$\mathbb{E}[g(X)] - \mathbb{E}[g(Y)] = \frac{1}{2} \int_0^1 \sum_{1 \leq i,j, \leq n} (\mathbb{E}[X_i X_j] - \mathbb{E}[Y_i Y_j]) \mathbb{E}[\frac{\partial g}{\partial x_i \partial x_j}(\sqrt{1-u}X + \sqrt{u}Y)] du \tag{14}$$

The result of Lemma 5 is valid when the second derivatives of $g$ exist only in the distribution sense (e.g. Dirac mass).

*Proof.* We define the function $G$ on $\mathbb{R}$ by

$$G(t) = \mathbb{E}[g(tX + \sqrt{1-t^2}Y)]$$

we have that

$$G'(t) = \sum_{i=1}^{n} \mathbb{E}[(X_i - \frac{t}{\sqrt{1-t^2}}Y_i)\frac{\partial g}{\partial x_i}(tX + \sqrt{1-t^2}Y)]$$

Moreover, it is easy to see that for any random vector $Z$ in $\mathbb{R}^n$ we have $\mathbb{E}[Z_i g(Z)] = \sum_{j=1}^{n} cov(X_i, X_j)\mathbb{E}[\frac{\partial g}{\partial x_j}(Z)]$, this yields

$$G'(t) = t \sum_{i,j=1}^{n} (\mathbb{E}[X_i X_j] - \mathbb{E}[Y_i Y_j])\mathbb{E}[\frac{\partial g}{\partial x_i \partial x_j}(tX + \sqrt{1-t^2}Y)]$$

We conclude by integrating $G'(t)$ between 0 and 1. $\qquad\square$

Now let $\mathcal{D} = \{(x_i, z_i) : 1 \leq i \leq N\}$ be the datapoints. Using the same notations as in the previous chapter, let $y^L(x_i)$ denotes on the neurons of $l^{th}$ layer (the neurons are iid). Assume $c^1_{x_i, x_j} \geq 0$ for all $i, j \in [1, N]$ (this is almost always the case, but in general, we can re-scale the input data to satisfy this assumption). We have the following result

**Lemma 6.** *Let $\phi$ be a non ReLU-like activation function and $(\sigma_b, \sigma_w) \notin EOC$. Then there exists $(\sigma_{b,EOC}, \sigma_{w,EOC}) \in EOC$ such that for any function $g \in \mathcal{D}^2$ such that for all $i, j \in [[1, N]], \frac{\partial g}{\partial x_i \partial x_j} \geq 0$, there exist $\beta > 0, \zeta_N > 0$ such that*

$$\mathbb{E}[g(y^L_{eoc}(X))] \leq \mathbb{E}[g(y^L_{ord}(X))] - \frac{\zeta_N}{L^\beta}$$

*Proof.* Let $(\sigma_b, \sigma_w) \notin EOC$ and $q$ be the corresponding limiting variance. From the previous chapter, it is easy to see that there exists $\sigma_0 > 0$ such that $\sigma_0^2 + \frac{\mathbb{E}[\phi(\sqrt{q}Z)^2]}{\mathbb{E}[\phi'(\sqrt{q}Z)^2]} = q$. Let $(\sigma_{b,EOC}, \sigma_{w,EOC}) = (\sigma_0, 1/\sqrt{\mathbb{E}[\phi'(\sqrt{q}Z)^2]}) \in EOC$. There exists a constant $\kappa > 0$ (independent of $N$) such that for all $i, j \in [[1, N]], \mathbb{E}[y^L_{eoc}(X_i)y^L_{eoc}(X_j)] \leq \mathbb{E}[y^L_{ord}(X_i)y^L_{ord}(X_j)] - \kappa L^{-\beta}$ where $\beta = 1$ for a smooth activation functions in $\mathcal{S}$ and $\beta = 2$ for ReLU-like activation functions (see Hayou et al. (2019)). Let $\lambda_N = \inf_{i,j} \inf_{u \in [0,1]} \mathbb{E}[\frac{\partial g}{\partial x_i \partial x_j}(\sqrt{1-u}X + \sqrt{u}Y)]$. Using Lemma 5, we have

$$\mathbb{E}[y^L_{eoc}(X_i)y^L_{eoc}(X_j)] \leq \mathbb{E}[y^L_{eoc}(X_i)y^L_{ord}(X_j)] - \frac{1}{2}\kappa N^2 \lambda_N L^{-\beta}$$

$\qquad\square$

As a simple application, using the function $g(X) = \prod_{i=1}^{N} 1_{x_i \leq t_i}$, we have the following

**Lemma 7.** *Let $t_1, t_2, ..., t_N \in \mathbb{R}$. Then there exist $\beta > 0, \zeta_N > 0$ such that*

$$\mathbb{P}(y^L_{eoc}(x_1) \leq t_1, ..., y^L_{eoc}(x_N) \leq t_N) \leq \mathbb{P}(y^L_{ord}(x_1) \leq t_1, ..., y^L_{ord}(x_N) \leq t_N) - \frac{\zeta_N}{L^\beta}$$

as a Corollary, we have a generalized form of Slepian's Lemma for $y^L_{eoc}(X)$ and $y^L_{ord}(X)$.

**Corollary 1** (Max Range and Min Range).

$$\mathbb{P}(\max_i y^L_{eoc}(x_i) \geq t) \geq \mathbb{P}(\max_i y^L_{ord}(x_i) \geq t) + \frac{\zeta_N}{L^\beta}$$

*and*

$$\mathbb{P}(\min_i y^L_{eoc}(x_i) \leq t) \geq \mathbb{P}(\max_i y^L_{ord}(x_i) \leq t) + \frac{\zeta_N}{L^\beta}$$

where $\beta = 1$ for activation functions of type $\mathcal{S}$ and $\beta = 2$ for ReLU-like activation functions.

# D   Training with SGD instead of GD

In this section, we extend the results of the NTK to the case of SGD. We use an approximation of the SGD dynamics by a diffusion process. We assume implicitly the existence of the triplet $(\Omega, \mathbb{P}, \mathcal{F})$ where $\Omega$ is the probability space, $\mathbb{P}$ is a probability measure on $\Omega$, and $\mathcal{F}$ is the natural filtration of the Brownian motion. Under boundedness conditions, when using SGD, the gradient update can be seen as a GD with a Gaussian noise (Hu et al., 2018; Li et al., 2017). More precisely, let $S = o(N)$ be the batchsize. The SGD update is given by

$$\hat{\theta}_{t+1} = \hat{\theta}_t - \eta \nabla_\theta \mathcal{L}^{(S)}(\hat{\theta}_t), \tag{15}$$

where $\mathcal{L}^{(S)} = \frac{1}{S} \sum_{i=1}^{S} \ell(f(\tilde{x}_i, \theta), \tilde{y}_i)$ where $(\tilde{x}_i, \tilde{y}_i)_{1 \le i \le S}$ is a randomly selected batch of size $S$. Then for all $\theta$

$$\nabla_\theta \mathcal{L}^{(S)}(\theta) - \nabla_\theta \mathcal{L}(\theta) = \sum_i \frac{Z_i(S)}{S}(\nabla_\theta \ell(f_\theta(x_i), y_i) - E_0(\theta)) - \sum_{i=1}^{N} \frac{(\nabla_\theta \ell(f_\theta(x_i), y_i) - E_0(\theta))}{N}$$

where $Z_i(S) = 1$ if observation $i$ belongs to the batch $(\tilde{x}_j, \tilde{y}_j), j \le S$ and equals 0 otherwise and $E_0(\theta) = \mathbb{E}_0 \nabla_\theta \ell(f(X_1, \theta), Y_1)$. We have

$$\mathrm{tr}\left[\mathrm{Cov}\left(\sum_{i=1}^{N} \frac{(\nabla_\theta \ell(f_\theta(x_i), y_i) - E_0(\theta))}{N}\right)\right] = \sum_{l=1}^{p} \frac{\mathrm{Var}\left(\partial \ell(f_\theta(X_1), Y_1)/\partial \theta_l\right)}{N}.$$

So that if $S = o(N)$ and if

$$\mathrm{tr}\left(\mathrm{Cov}\left(\nabla_\theta \ell(f(X_1, \theta), Y_1)\right)\right) = o(S)$$

where $\mathrm{Cov}(\cdot)$ denotes the covariance matrix under $\mathbb{P}_0$. Then

$$\nabla_\theta \mathcal{L}^{(S)}(\theta) - \nabla_\theta \mathcal{L}(\theta) = \frac{Z_S(\theta)}{\sqrt{S}} + o_{P_0}(S^{-1/2})$$

where $Z_S(\theta)$ converges in distribution (as $S$ goes to infinity) to a Gaussian random vector with covariance matrix $\Sigma(\theta) = \mathrm{Cov}\left(\nabla_\theta \ell(f(X_1, \theta), Y_1)\right)$ and we have, neglecting the term $o_{P_0}(S^{-1/2})$,

$$\hat{\theta}_{t+1} = \hat{\theta}_t - \eta \nabla_\theta \mathcal{L}(\hat{\theta}_t) + \frac{\eta}{\sqrt{S}} Z(\theta_t). \tag{16}$$

We can in particular bound the difference between equation 16 and the continuous time SDE approximation (see also Hu et al. (2018) and Li et al. (2017))

$$d\theta_t = -\nabla_\theta \mathcal{L}(\theta_t)dt + \sqrt{\frac{\eta}{S}} \Sigma(\theta_t)^{\frac{1}{2}} dW_t. \tag{17}$$

SGD updates can therefore be seen as a discretization of the previous SDE with time step $\Delta = \eta$, and where $\Sigma(\theta_t)^{\frac{1}{2}}$ is the square-root matrix of $\Sigma(\theta_t) = \mathrm{Cov}\left(\nabla_\theta \ell(f(X_1, \theta_t), Y_1)\right)$ and $(W_t)_{t \ge 0}$ a standard Brownian motion.

Since the dynamics of $\theta_t$ are described by an SDE, the dynamics of $f_t$ can also be described by an SDE which can be obtained from Itô's lemma.

**Proposition 4.** *Under the dynamics of the SDE equation 17, the vector $f_t(\mathcal{X})$ is the solution of the following SDE*

$$df_t(\mathcal{X}) = [-\frac{1}{N} K_{\theta_t}^L(\mathcal{X}, \mathcal{X}) \nabla_z \ell(f_t(\mathcal{X}), Y) + \frac{1}{2}\frac{\eta}{S}\Gamma_t(\mathcal{X})]dt + \sqrt{\frac{\eta}{S}} \nabla_\theta f(\mathcal{X}, \theta_t) \Sigma(\theta_t)^{\frac{1}{2}} dW_t \tag{18}$$

*where $\Gamma_t(\mathcal{X})$ is the concatenated vector of $(\Gamma_t(x) = (\mathrm{Tr}(\Sigma(\theta_t)^{\frac{1}{2}} \nabla_2 f_i(x, \theta_t) \Sigma(\theta_t)^{\frac{1}{2}}))_{1 \le i \le o})_{x \in \mathcal{X}}$ and $\nabla_2 f_i(x, \theta)$ is the Hessian of $f_i$ ($i^{th}$ component of $f$) with respect to $\theta$.*

*Proof.* Since $\theta_t$ is a diffusion process, we can use Itô's lemma to deduce how the randomness propagates to $f_t$. We denote by $f_{t,i}$ the $i^{th}$ coordinate of $f_t$, i.e., for an input $x$, $f_t(x) = (f_{t,i}(x))_{1 \leq i \leq k}$. Let $i \in 1, ..., k$, we have

$$df_{t,i}(x) = \nabla_\theta f_i(x, \theta_t)d\theta_t + \frac{1}{2}\frac{\eta}{S}\text{Tr}(\Sigma(\theta_t)^{\frac{1}{2}}\nabla_2 f_i(x, \theta_t)\Sigma(\theta_t)^{\frac{1}{2}})dt$$

$$= [-\nabla_\theta f_{t,i}(x)\nabla_\theta f_t(\mathcal{X})\nabla_z \ell(f_t(\mathcal{X}), Y) + \frac{1}{2}\frac{\eta}{S}\text{Tr}(\Sigma(\theta_t)^{\frac{1}{2}}\nabla_2 f_i(x, \theta_t)\Sigma(\theta_t)^{\frac{1}{2}})]dt$$

$$+ \sqrt{\frac{\eta}{S}}\nabla_\theta f_i(x, \theta_t)\Sigma(\theta_t)^{\frac{1}{2}}dW_t$$

where $\nabla_2 f_i(x, \theta_t)$ is the hessian of $f_i$ with respect to $\theta$. Aggregating these equations with respect to $i$ yields

$$df_t(x) = [-\frac{1}{N}\nabla_\theta f_t(x)\nabla_\theta f_t(\mathcal{X})\nabla_z \ell(f_t(\mathcal{X}), Y) + \frac{1}{2}\frac{\eta}{S}\Gamma_t(x)]dt + \sqrt{\frac{\eta}{S}}\nabla_\theta f(x, \theta_t)\Sigma(\theta_t)^{\frac{1}{2}}dW_t$$

where $\Gamma_t(x) = (\text{Tr}(\Sigma(\theta_t)^{\frac{1}{2}}\nabla_2 f_i(x, \theta_t)\Sigma(\theta_t)^{\frac{1}{2}}))_{1 \leq i \leq k}$.

Therefore, the dynamics of the vector $f_t(\mathcal{X})$ is given by

$$df_t(\mathcal{X}) = [-\frac{1}{N}K_{\theta t}(\mathcal{X}, \mathcal{X})\nabla_z \ell(f_t(\mathcal{X}), Y) + \frac{1}{2}\frac{\eta}{S}\Gamma_t(\mathcal{X})]dt + \sqrt{\frac{\eta}{S}}\nabla_\theta f(\mathcal{X}, \theta_t)\Sigma(\theta_t)^{\frac{1}{2}}dW_t$$

where $\Gamma_t(\mathcal{X})$ is the concatenated vector of $(\Gamma_t(x))_{x \in \mathcal{X}}$. $\qquad\square$

With the quadratic loss $\ell(z, y) = \frac{1}{2}||z - y||^2$, the SDE equation 18 is equivalent to

$$df_t(\mathcal{X}) = [-\frac{1}{N}K_{\theta_t}^L(\mathcal{X}, \mathcal{X})(f_t(\mathcal{X}) - \mathcal{Y}) + \frac{1}{2}\frac{\eta}{S}\Gamma_t(\mathcal{X})]dt + \sqrt{\frac{\eta}{S}}\nabla_\theta f(\mathcal{X}, \theta_t)\Sigma(\theta_t)^{\frac{1}{2}}dW_t. \quad (19)$$

This is an Ornstein-Uhlenbeck process (mean-reverting process) with time dependent parameters. The additional term $\Gamma_t$ is due to the randomness of the mini-batch, it can be seen as a regularization term and could partly explain why SGD gives better generalization errors compared to GD (Kubo et al. (2019), Lei et al. (2018)).

**Dynamics of $f_t$ for wide FeedForward neural networks :**
In the case of a fully connected feedforward neural network (FFNN hereafter) of depth $L$ and widths $n_1, n_2, ..., n_L$, Jacot et al. (2018) proved that, with GD, the kernel $K_{\theta_t}^L$ converges to a kernel $K^L$ that depends only on $L$ (number of layers) for all $t < T$ when $n_1, n_2, ..., n_l \to \infty$, where $T$ is an upper bound on the training time, under the technical assumption $\int_0^T ||\nabla_z \ell(f_t(\mathcal{X}, \mathcal{Y}))||_2 dt < \infty$ almost surely with respect to the initialization. For SGD, we assume that the convergence result of the NTK holds true as well, this is illustrated empirically in figure 3 but we leave the theoretical proof for future work. With this approximation, the dynamics of $f_t(\mathcal{X})$ for wide networks is given by

$$df_t(\mathcal{X}) = -\frac{1}{N}\hat{K}^L(f_t(\mathcal{X}) - M_t)dt + \sqrt{\frac{\eta}{S}}\nabla_\theta f(\mathcal{X}, \theta_t)\Sigma(\theta_t)^{\frac{1}{2}}dW_t,$$

where $\hat{K}^L = K^L(\mathcal{X}, \mathcal{X})$ and $M_t = Y - \frac{\eta N}{2S}(\hat{K}^L)^{-1}\Gamma_t(\mathcal{X})$. This is an Ornstein–Uhlenbeck process whose closed-form expression is given by

$$f_t(\mathcal{X}) = e^{-\frac{t}{N}\hat{K}^L}f_0(\mathcal{X}) + (I - e^{-\frac{t}{N}\hat{K}^L})\mathcal{Y} + A_t(\mathcal{X}) \quad (20)$$

where $A_t(\mathcal{X}) = -\frac{\eta}{2S}\int_0^t e^{-\frac{t-s}{N}\hat{K}^L}\Gamma_s(\mathcal{X})ds + \sqrt{\frac{\eta}{S}}\int_0^t e^{-\frac{t-s}{N}\hat{K}^L}\nabla_\theta f(\mathcal{X}, \theta_s)\Sigma(\theta_s)^{\frac{1}{2}}dW_s$; see supplementary material for the proof. So for any (test) input $x \in \mathbb{R}^d$, we have

$$f_t(x) = f_0(x) + K^L(x, \mathcal{X})(\hat{K}^L)^{-1}(I - e^{-\frac{t-s}{N}\hat{K}^L})(\mathcal{Y} - f_0(\mathcal{X})) + Z_t(x) + R_t(x), \quad (21)$$

where $R_t(x) = \sqrt{\frac{\eta}{S}}\int_0^t[K^L(x, \mathcal{X})(\hat{K}^L)^{-1}(e^{-\frac{t}{N}\hat{K}^L} - I)\nabla_\theta f(\mathcal{X}, \theta_s) + \nabla_\theta f(x, \theta_s)]\Sigma(\theta_s)^{\frac{1}{2}}dW_s$ and $Z_t(x) = \frac{\eta}{2S}\left[\int_0^t \Gamma_s(x)ds + \int_0^t K(x, \mathcal{X})(\hat{K}^L)^{-1}(I - e^{-\frac{(t-s)}{N}\hat{K}^L})\Gamma_s(\mathcal{X})ds\right]$.

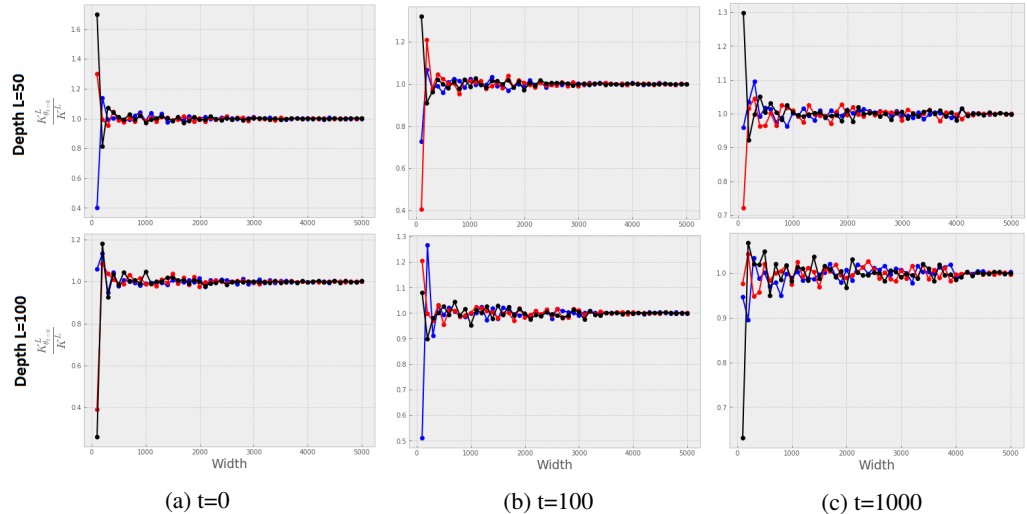

(a) t=0  (b) t=100  (c) t=1000

Figure 3: Ratio $K_{\theta_t}^L/K^L$ for three randomly selected pairs from MNIST dataset as a function of width for three training times $t = 0$, $t = 100$ and $t = 1000$ (training time is measured by SGD updates)

*Proof.* Using the approximation of the NTK by $K^L$ as $n_1, n_2, ..., n_L \to \infty$, the dynamics of $f_t(\mathcal{X})$ for wide networks are given by

$$df_t(\mathcal{X}) = -\frac{1}{N}\hat{K}^L(f_t(\mathcal{X}) - M_t)dt + \sqrt{\frac{\eta}{S}}\nabla_\theta f(\mathcal{X}, \theta_t)\Sigma(\theta_t)^{\frac{1}{2}}dW_t,$$

To solve it, we use the change of variable $A_t = e^{\frac{t}{N}\hat{K}^L}f_t(\mathcal{X})$. Using Ito's lemma, we have

$$dA_t = \frac{1}{N}\hat{K}^L A_t dt + e^{\frac{t}{N}\hat{K}^L}df_t(\mathcal{X})$$
$$= \frac{1}{N}\hat{K}^L e^{\frac{t}{N}\hat{K}^L}M_t dt + \sqrt{\frac{\eta}{S}}e^{\frac{t}{N}\hat{K}^L}\nabla_\theta f(\mathcal{X}, \theta_t)\Sigma(\theta_t)^{\frac{1}{2}}dW_t$$

By integrating, we conclude that

$$f_t(\mathcal{X}) = e^{-\frac{t}{N}\hat{K}^L}f_0(\mathcal{X}) + \frac{1}{N}\int_0^t \hat{K}^L e^{\frac{-(t-s)}{N}\hat{K}^L}M_s ds + \sqrt{\frac{\eta}{S}}\int_0^t e^{-\frac{t-s}{N}\hat{K}^L}\nabla_\theta f(\mathcal{X}, \theta_s)\Sigma(\theta_s)^{\frac{1}{2}}dW_s$$

we conclude for $f_t(\mathcal{X})$ using the fact that $\frac{1}{N}\int_0^t \hat{K}^L e^{\frac{-(t-s)}{N}\hat{K}^L}M_s ds = (I - e^{-\frac{t}{N}\hat{K}^L})\mathcal{Y} - \frac{\eta}{2S}\int_0^t e^{\frac{-(t-s)}{N}\hat{K}^L}\Gamma_s ds$.

Recall that for any input $x \in \mathbb{R}^d$,

$$df_t(x) = [-\frac{1}{N}K^L(x, \mathcal{X})(f_t(\mathcal{X}) - Y) + \frac{1}{2}\frac{\eta}{S}\Gamma_t(x)]dt + \sqrt{\frac{\eta}{S}}\nabla_\theta f(x, \theta_t)\Sigma(\theta_t)^{\frac{1}{2}}dW_t$$

To prove the expression of $f_t(x)$ for general $x \in \mathbb{R}^d$, we substitute $f_t(\mathcal{X})$ by its value in the SDE of $f_t(x)$ and integrate.

□

# E  ADDITIONAL EXPERIMENTS : IMPACT OF THE PARAMETRIZATION

Figure 4 shows the impact of different parametrizations on the training dynamics. It shows that both standard parametrization and NTK parametrization have similar behaviour on different initialization regimes.

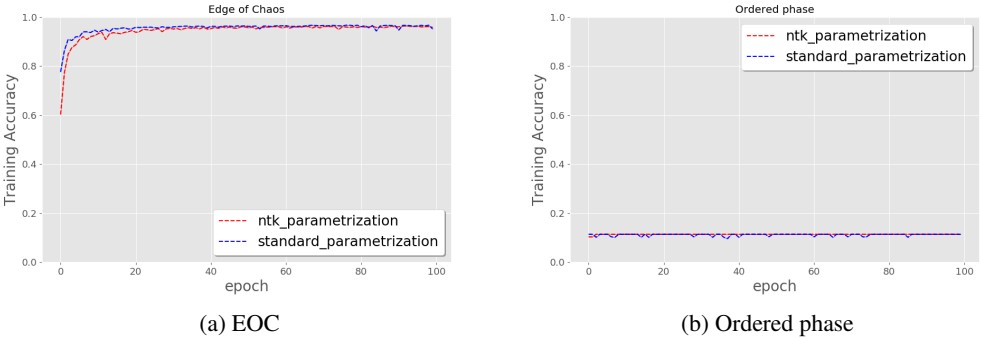

(a) EOC         (b) Ordered phase

Figure 4: Impact of NTK parametrization on the Training

