# OpenReview forum: "Mean-field Behaviour of Neural Tangent Kernel for Deep Neural Networks"
_ICLR.cc/2020/Conference — Reject_

### Official Review · AnonReviewer2 · 2019-10-15
**Official Blind Review #2**

**Rating:** 6

**Review:**

This paper discussed the property of the NTK with the increasing depth L with the help of the Edge of Chaos initialization. The authors show that if deep neural networks are not properly initialized, the NTK can have a large condition number, which leads to the poor performance of training and generalization. Moreover, the authors also introduce the conditions that make the neural network trainable by decreasing the convergence rate to a nearly constant kernel w.r.t the depth L by using the specific Edge of Chaos initialization as well as different activations and use residual connections. Experiment results show that the theoretical results are well aligned with the practice.

Detailed Comments:
1. At the paragraph after Lemma 1, the authors claimed that f_t(x) is entirely given by f_0(x), which means a generalization error of order O(1). This is a little inaccurate I think, as NTK can only characterize the training process and does not directly indicate generalization. Also, as t to infinity, the coefficient of initialization f_0(x) can be arbitrary small and thus the f_t(x) is entirely decided by initialization is not accurate as well. To analyze generalization with NTK, we need a little more, see [1]. This is not some core issue, but I think the authors should make the description more accurate.
2. The authors should explain what richer limiting NTK means at the end of the Sec 3. It is unclear how the convergence rate of the NTK related to the richer limiting NTK.
3. A typo in the first paragraph in Sec 4, ouerselvs to ourselves.
4. In definition 1, the second-order derivative of \phi is defined via the indicator function 1_{A_i}? It’s better to use another notation like \mathbf{1} or \mathbb{1} to make it more clear.
5. I think for clarity the authors should give a formal definition of ANTK, or if it is unnecessary, better directly use the original NTK. Also, it is better to show the invertibility of the NTK is equivalent to ANTK more formally before Proposition 2. This conclusion is not so straightforward at the first view. Meanwhile, I think the invertibility of ANTK can just indicate a bad condition number of NTK? I don’t think it directly related to the invertibility of NTK.
6. There are several typos in the appendix. Please go through the appendix and fix them.
7. In the proof of Lemma 3 and Lemma 4, I don’t get why |a_l| < l + |a_0|. It may depends on the property of q? For sufficiently large q, if a_0=0, a_1 = q + O(1) which is not necessarily smaller than l=1? But to get a_l/l bounded, it is not necessary to use this. Also, I think it’s better not omit the constant in the dynamic system of Lemma 3.
8. Better use the Stirling’s approximation rather than k to infinity in the calculation of k!/(k-\alpha-1)!
9. The lemma number of Appendix C is in a chaos. And I don’t quite get the idea behinds Appendix C. Is it relevant to the contents in the main text?
10. I also feel the content in Appendix D is unnecessary for this paper.

Overall, this paper is interesting and gives a unified perspective on the recently developed NTK and Edge of Chaos initialization. It also sheds light on the impact of different activation function on NTK by generalizing the results of [2]. I feel this paper can help the communities have more understanding on the properties of deep neural networks. However, this paper can be better if the authors can polish their paper and reorganize the appendix part.

[1] Arora, Sanjeev, et al. "Fine-Grained Analysis of Optimization and Generalization for Overparameterized Two-Layer Neural Networks." International Conference on Machine Learning. 2019.
[2] Hayou, S., Doucet, A. & Rousseau, J.. (2019). On the Impact of the Activation function on Deep Neural Networks Training. Proceedings of the 36th International Conference on Machine Learning, in PMLR 97:2672-2680


**Experience Assessment:**

I have read many papers in this area.

**Review Assessment: Checking Correctness Of Derivations And Theory:**

I assessed the sensibility of the derivations and theory.

**Review Assessment: Checking Correctness Of Experiments:**

I assessed the sensibility of the experiments.

**Review Assessment: Thoroughness In Paper Reading:**

I read the paper thoroughly.

---

> ### Author Response · Authors · 2019-11-10
> **Re : Official Blind Review #2**
>
> We thank Reviewer 2 for the detailed and insightful comments. We address them hereafter.
>
> 1) “At the paragraph after Lemma 1, the authors claimed that f_t(x) is entirely given by f_0(x), which means a generalization error of order O(1). This is a little inaccurate I think, as NTK can only characterize the training process and does not directly indicate generalization. Also, as t to infinity, the coefficient of initialization f_0(x) can be arbitrary small and thus the f_t(x) is entirely decided by initialization is not accurate as well. To analyze generalization with NTK, we need a little more, see [1]. This is not some core issue, but I think the authors should make the description more accurate. ”
> We have clarified the text in the revised version of the paper. By ‘generalization error’, we were only referring to the generalization of the linear model given by equation (9) and not the generalization of the neural network. This linear model can be a good approximation of the true dynamics of a neural network (see Lee et al (2019) for example).  As t grows to infinity in equation (9), assuming the NTK remains invertible, we end up with a formula of f_t(x) that depends only on f_0 and the kernel (which we assume to have constant covariances), therefore f_t(x) is entirely determined by f_0(x).
> Moreover, as Reviewer 2 mentioned, discussing the generalization error of a neural network from an NTK point of view would require a different technical analysis and is beyond the scope of this paper.
>
>
> 2) “The authors should explain what richer limiting NTK means at the end of the Sec 3. It is unclear how the convergence rate of the NTK related to the richer limiting NTK. ”
> By ‘richer’ kernel we meant that with smooth activation functions, the NTK loses less information compared to when ReLU-like activations are used as the depth grows to infinity. We have rewritten this part of the paper.
>
> 3) “I think for clarity the authors should give a formal definition of ANTK, or if it is unnecessary, better directly use the original NTK. Also, it is better to show the invertibility of the NTK is equivalent to ANTK more formally before Proposition 2. This conclusion is not so straightforward at the first view. Meanwhile, I think the invertibility of ANTK can just indicate a bad condition number of NTK? I don’t think it directly related to the invertibility of NTK. ”
> The ANTK is just a scaled version of the NTK (ANTK = NTK / L). So the invertibility of the NTK is equivalent to that of the ANTK, and the condition numbers of both the ANTK and the NTK are the same (here we define the condition number to be largest_eigenvalue / smallest_eigenvalue). We have clarified the definition of the ANTK in the revised version of the paper.
>
>
> 4) “In the proof of Lemma 3 and Lemma 4, I don’t get why |a_l| < l + |a_0|. It may depends on the property of q? For sufficiently large q, if a_0=0, a_1 = q + O(1) which is not necessarily smaller than l=1? But to get a_l/l bounded, it is not necessary to use this. Also, I think it’s better not omit the constant in the dynamic system of Lemma 3. ”
> Thank you for pointing this out. Indeed, it should be “constant * l” in the right hand part. We have corrected this error in the revised version.
>
>
> 5) “Better use the Stirling’s approximation rather than k to infinity in the calculation of k!/(k-\alpha-1)! ”
> We used an easy equivalent since it is a sum of a divergent series with positive terms, but Stirling’s formula would work too.
>
> 6)“The lemma number of Appendix C is in a chaos. And I don’t quite get the idea behinds Appendix C. Is it relevant to the contents in the main text?
> ”
> Thank you, we have fixed the numbering in the revised version. Results in Appendix C were introduced to confirm the impact of the EOC on the output function of the neural network. We agree that this is not directly linked to the NTK. However, as the NTK is a covariance matrix of the Gradient of the output function, we believe it is interesting to understand the behaviour of the output function for different initializations.
>
>
> 7)“I also feel the content in Appendix D is unnecessary for this paper. ”
> So far, the NTK has only been studied with GD. The randomness of SGD makes it difficult to understand the limiting behaviour of the NTK in the infinite width limit, and we might even end up with a random NTK in some cases. We introduced Appendix D to shed light on this problem and pave the way for future projects in this direction.
>
>
> [1] Lee et al.(2019) “Wide neural networks of any depth evolve as linear models under gradient descent.”

---

> > ### Comment · AnonReviewer2 · 2019-11-14
> > **Thanks for your clarification.**
> >
> > I feel the new version is much better than the original one. I will vote for acceptance and I hope the authors can continuously polish the paper so that these interesting results can be easily accepted by the potential audiences.

---

### Official Review · AnonReviewer1 · 2019-10-23
**Official Blind Review #1**

**Rating:** 6

**Review:**

This paper studies the role of weight/bias variance of neural network’s trainability via analyzing large depth behaviour of Neural Tangent Kernels(NTK).

Recently NTK has been a popular topic of study in theoretical deep learning as it describes exact gradient descent dynamics of infinitely wide networks. Original NTK paper (Jacot et al. (2018)) and other follow up papers often gloss over role of weight/bias scales whereas in the setting of signal propagation or NNGP covariance, Schoenholz et al. (2017), Lee et al. (2018), Novak et al. (2019), Hayou et al. (2019) have shown understanding initialization scale is quite important as networks becomes deeper. This paper brings those analysis for NTK and discovers few interesting results.

The good weight/bias initialization scale that propagates signal for very deep network is denoted edge of chaos (EOC). The authors show that 1) for fully connected feed forward networks, outside initialization edge of chaos (EOC) the NTK converges exponentially to constant kernel. This indicates non-trainability. 2) With EOC initialization the convergence is polynomial and the NTK remains invertible for very large depth implying trainability. 3) Certain class of activation function (denoted class S, including ELU/Tanh/Swish) it has even slower convergence(O(log(L)/L)) compare to ReLU (O(1/L)). 4) Residual FC networks NTK has polynomial convergence for all weight and bias variance.

In terms of novelty, the paper is combining existing techniques and objects to study large depth behaviour of NTK. However the contribution of the paper is interesting and worth the ICLR audience to know about, especially with the current surge of interest in NTK.

One main weakness is that the experiments are weak and have a very weak connection to the early part of the paper. Section 5.1 is direct convergence comparison, which provides fair evidence. In section 5.2, I’m not certain whether experiments displayed connects to asymptotic NTK analysis. First of all, in order to obtain NTK in the infinite width limit, one has to use `NTK parameterization’ where one scales out 1/sqrt(fan_in) in network definition and not in weight initialization. It seems (by mention of He/Glorot init, and choice of learning rate O(1e-3/1e-4)) the authors experimented with standard parameterization. In this case the connection to very deep behaviour of NTK is not straightforward to actual network training since the training dynamics will be different.  I suggest authors try experiments in NTK parameterization and see if results are similar.


Few comments:
It should be emphasized that the infinite width is taken first before taking infinite depth limit. There are subtle effects when depth/width are taken to infinity at the same time. The analysis on asymptotic behaviour of NTK at infinite depth is only valid after width is taken to infinity.

First sentence of the abstract is too strong in the sense that NTK has limitations. NTK certainly does not work for any kind of network, so I suggest authors to down tone the sentence.

P4 paragraph after Lemma1 `'K^L(x, x’) = cte’ is a typo?

In section 5.2, the authors’ claim full batch GD is practically impossible. One could accumulate gradients over minibatch to simulate full batch GD.

P13 typo in last paragraph, `  NTKl'

EDITS POST AUTHOR RESPONSE:
Regarding point 5) I was referring to solving the memory problem. In practice, if one is sampling without replacement, there is no randomness accumulating gradient of the full epoch.

I also thank the authors for the clarification. My score still remains the same.

**Experience Assessment:**

I have published in this field for several years.

**Review Assessment: Checking Correctness Of Derivations And Theory:**

I assessed the sensibility of the derivations and theory.

**Review Assessment: Checking Correctness Of Experiments:**

I assessed the sensibility of the experiments.

**Review Assessment: Thoroughness In Paper Reading:**

I read the paper at least twice and used my best judgement in assessing the paper.

---

> ### Author Response · Authors · 2019-11-10
> **Re : Official Blind Review #1**
>
> We thank Reviewer 1 for the detailed and insightful feedback. We address them hereafter.
>
> 1) “One main weakness is that the experiments are weak and have a very weak connection to the early part of the paper. Section 5.1 is direct convergence comparison, which provides fair evidence. In section 5.2, I’m not certain whether experiments displayed connects to asymptotic NTK analysis... ”
>
> We agree that the dynamics might be slightly different. However, in practice both parametrizations give similar results in terms of Trainability and Training Speed (which are our focus in this paper) since they both have the same EOC (same EOC curve  in (\sigma_b, \sigma_w) plane). More precisely, the NTK is the same for both parametrization at initialization, and in practice we use small learning rates for standard parametrization (1e-3) compared to the learning rates used with NTK parametrization (to compensate for the scaling), we believe that the resulting training dynamics should not change much, unless we use small learning rates for the NTK parametrization which would result in very slow training. We added in the appendix (Appendix E) an example of training a network of depth 100 and width 100 with both parametrizations with an EOC initialization (we used ELU in this example), it shows that both parametrizations have similar behaviour on the EOC. This is also true on the Ordered/Chaotic phase where the model is stuck at the random classifier accuracy ~10% for both parametrizations.  We will add more experimental results with NTK Parametrization in the appendix of the final version of the paper.
>
> 2) “It should be emphasized that the infinite width is taken first before taking infinite depth limit. There are subtle effects when depth/width are taken to infinity at the same time. The analysis on asymptotic behaviour of NTK at infinite depth is only valid after width is taken to infinity. “
> We have mentioned that we only deal with the infinite width networks in “Motivation and Related work”. We have changed it in the revised version to make it clearer that the width is considered infinite before taking the limit of large depth.
>
>
> 3)”First sentence of the abstract is too strong in the sense that NTK has limitations. NTK certainly does not work for any kind of network, so I suggest authors to down tone the sentence. ”
> In the first sentence of the abstract, we are only referring to the exact NTK and not Mean-Field NTK. The Exact NTK is just another formulation of the Gradient Descent optimization problem as a Gradient in the Functional space, the exact NTK changes with training time t. We agree that when considering the Mean-Field NTK, that sentence is no longer correct.
>
>
> 4) “P4 paragraph after Lemma1 `'K^L(x, x’) = cte’ is a typo? ” and “P13 typo in last paragraph, ` NTKl' ”
> Thank you. We have fixed the typos in the revised version
>
>
> 5) “In section 5.2, the authors’ claim full batch GD is practically impossible. One could accumulate gradients over minibatch to simulate full batch GD. ” This is a very interesting suggestion.  Just to make sure we understand correctly what you mean: by simulating the GD using minibatch gradients, we will still have some randomness, isn’t that equivalent to SGD with bigger batchsize ? Or are you suggesting this could potentially solve the memory cost problem ?

---

### Official Review · AnonReviewer3 · 2019-10-23
**Official Blind Review #3**

**Rating:** 3

**Review:**

The paper studies the limiting behavior of neural tangent kernels when the depth grows to infinity. They show that the obtained limit kernels are trivial (a constant) unless one uses 'edge of caos' initialization, in which case they are close to the identity. The authors compare the convergence for different activations, showing a slower convergence (hence better propagation) for some piecewise smooth activations compared to ReLU. For residual networks, the 'edge of caos' behavior is claimed to be in place regardless of the initialization.

The detailed characterization of these limiting kernels for different activations and initializations is interesting. Yet, the work is quite incremental and of limited significance, in that such EOC initialization was known to be important for controlling propagation of both activations and gradients with such networks, so it is no surprise that the NTK has similar properties, given that it basically consists of the sum of similar gradient covariances.

some comments:
- title: isn't the "mean-field behavior" already subsumed in the definition of NTK?
- contribution 4.: "more suitable" seems a bit strong if it's just a log(L) factor?
- after lemma 1: generalization may not provide a strong argument here unless further discussed: the constant kernel is obviously bad for learning anything, but the EOC limiting kernel is also pretty bad for predicting anything outside the training data
- section 3: typo "ourselves"
- prop 3: specify that phi is the relu
- the proof of prop 3 should be more detailed. Also, it is not obvious that you are giving the correct formula for the NTK of resnets
- Table 1: is it meaningful to compare performance with such low accuracies?

**Experience Assessment:**

I have read many papers in this area.

**Review Assessment: Checking Correctness Of Derivations And Theory:**

I assessed the sensibility of the derivations and theory.

**Review Assessment: Checking Correctness Of Experiments:**

I assessed the sensibility of the experiments.

**Review Assessment: Thoroughness In Paper Reading:**

I read the paper at least twice and used my best judgement in assessing the paper.

---

> ### Author Response · Authors · 2019-11-09
> **Re : Official Blind Review #3 (part 2)**
>
> (this is part 2 of the response, please read part 1 first)
>
> 6) “- Table 1: is it meaningful to compare performance with such low accuracies? ” The goal here is mainly trainability and training speed of neural networks, and we approach this problem from a NTK point of view. We want to confirm the findings by training neural networks with different activations on MNIST and CIFAR10. We compare accuracies after 10 epochs and 100 epochs of training to see whether the model is trainable first and which configuration is training faster. This table shows two facts: when the NTK converges exponentially fast to a constant non-invertible kernel (which is the case for Sigmoid and Softplus since they do not have an EOC), the training is impossible and the model is stuck at a low accuracy of ~10% which is the accuracy of a random classifier. The second fact is that, on the EOC, while for all activations (that have an EOC) the model is trainable, smooth activation functions like ELU and Tanh perform better than ReLU, Leaky-ReLU and PReLU. This confirms the results of Proposition 2.
>
>
> [1] Lee et al.(2019) “Wide neural networks of any depth evolve as linear models under gradient descent.”
> [2] Schoenholz et al. (2017)  “Deep information propagation ”
> [3] Hayou et al. (2019) “On the impact of the activation function on deep neural networks training”

---

> ### Author Response · Authors · 2019-11-09
> **Re : Official Blind Review #3 (part 1)**
>
> We thank Reviewer 3 for the feedback. We address raised concerns hereafter
>
> 1) “the work is quite incremental and of limited significance... so it is no surprise that the NTK has similar properties, given that it basically consists of the sum of similar gradient covariances. ”
>
> The NTK has attracted recently the interest of many researchers to analyse neural networks. New training methods based on it have also appeared; e.g. Lee et al. (2019) used a linear approximation of the training dynamics based on the NTK and showed that for different architectures, this simple dynamics gives similar results compared to standard GD training. However, these experiments were only successful for shallow neural networks. Here, we explain why the linear system approximation does not work for deep neural networks: it is mainly because of the initialization and the activation function. In this work, we bridge the gap between NTK and EOC. We show that any NTK-based training method for DNNs should use a Smooth Activation Function from the class S and the network should be initialized on the EOC. We believe these results better help understanding the training of DNNs, especially with NTK based training methods.
>
> We agree that the limiting behaviour of the NTK on the ordered phase were to be expected, given the results of Schoenholz et al. (2017) and Hayou et al. (2019). However, the limiting behaviour of the NTK on the EOC is more surprising. It is true that the EOC was already proven to avoid gradient variance from vanishing/exploding, but covariance backpropagation is different and we may have some examples where the covariance vanishes and the variance remains constant. Moreover, the Mean-Field theory of Gradient backpropagation (Schoenholz et al. (2017) and Hayou et al. (2019)) only shows that the variance of the gradient does not vanish/explode ‘exponentially’ on the EOC, in that it has sub-exponential growth/vanishing rates. However, to the best of our knowledge, the exact convergence rates have never been established before. Here we not only prove the asymptotic degeneracy of the kernel but we also provide the exact convergence rates of the ANTK for two different types of activation functions: ReLU-like activations (ReLU, Leaky-ReLU) with a convergence rate of O(1/L) and smooth functions from class S (ELU, Tanh, Swish/SiLU …) with a convergence rate of O(log(L)/L).
> Moreover, we discuss the behaviour of the NTK for Residual Neural Networks. We show that, by adding these residual connections, we actually avoid any exponential convergence to the limiting kernel. This means that we ‘live’ on the EOC whatever being the initial parameters. To the best of our knowledge, this has never been proven before. This could potentially explain the good performance of Residual Networks. These results also suggest that residual networks might be more suitable for NTK based training methods.
>
>
> 2) “title: isn't the "mean-field behavior" already subsumed in the definition of NTK? ”
> The NTK can be defined for any architecture and any model without assuming an infinite width limit. By mean-field we want to emphasize the fact that we deal with infinite width networks as an approximation of wide neural networks.
>
> 3) “- contribution 4.: "more suitable" seems a bit strong if it's just a log(L) factor? ” we agree that asymptotically, the log(L) factor does not change much. However, for medium depths of around 100, we have log(L) ~ 5 , which means that ReLU DNN of 100 layers loses 5 time more information compared to other networks with smooth activations such as ELU. This could have a non-negligible impact on the training.
>
>
> 4) “- after lemma 1: generalization may not provide a strong argument here unless further discussed: the constant kernel is obviously bad for learning anything, but the EOC limiting kernel is also pretty bad for predicting anything outside the training data ”
> We agree that generalization needs further attention. The convergence to the limiting ANTK on the EOC is also poor for generalization, but we know at least that is it has a very slow convergence. This was meant to be a ‘necessary’ argument rather than ‘sufficient’ for the generalization of the model given by equation (9). We have clarified this in the revised version of the paper.
>
> 5) “- the proof of prop 3 should be more detailed. Also, it is not obvious that you are giving the correct formula for the NTK of resnets ” The only difference in the Residual NTK compared to FeedForward NTK is the residual term. We have clarified this in the revised version of the paper.

---

### Decision · Program_Chairs · 2019-12-19

**Decision:**

Reject

**Comment:**

This paper focuses on studying the impact of initialization and activation functions on the Neural Tangent Kernel (NTK) type analysis. The authors claim to make a connection between NTK and edge of chaos analysis. The reviewers had some concern about (1) impact of smooth activations "any NTK-based training method for DNNs should use a Smooth Activation Function from the class S and the network should be initialized on the EOC" (2) proofs of residual networks (3) and why mixing NTK with EOC is interesting. Some of these concerns were addressed in the response. I do share the reviewer concerns about (2). The authors need to give a clear proof. I think this combination of NTK and EOC could be interesting but needs to be better motivated. As a result I do not recommend publication.